# On the Out-of-Distribution Coverage of Combining Split Conformal Prediction and Bayesian Deep Learning

**Paul Scemama**                                          *pscemama@mitre.org*
*The MITRE Corporation*

**Ariel Kapusta**                                          *akapusta@mitre.org*
*The MITRE Corporation*

**Reviewed on OpenReview:** *https: // openreview. net/ forum? id= TySx8fsSSU*

## Abstract

Bayesian deep learning and conformal prediction are two methods that have been used to convey uncertainty and increase safety in machine learning systems. We focus on combining Bayesian deep learning with split conformal prediction and how the addition of conformal prediction affects out-of-distribution coverage that we would otherwise see; particularly in the case of multiclass image classification. We suggest that if the model is generally underconfident on the calibration dataset, then the resultant conformal sets may exhibit worse out-of-distribution coverage compared to simple predictive credible sets (i.e. not using conformal prediction). Conversely, if the model is overconfident on the calibration dataset, the use of conformal prediction may improve out-of-distribution coverage. In particular, we study the extent to which the addition of conformal prediction increases or decreases out-of-distribution coverage for a variety of inference techniques. In particular, (i) stochastic gradient descent, (ii) deep ensembles, (iii) mean-field variational inference, (iv) stochastic gradient Hamiltonian Monte Carlo, and (v) Laplace approximation. Our results suggest that the application of conformal prediction to different predictive deep learning methods can have significantly different consequences.

## 1 Introduction

Bayesian deep learning and conformal prediction are two paradigms that have been used to represent uncertainty and increase trust in machine learning systems. Bayesian deep learning attempts to endow deep learning models with the ability to represent predictive uncertainty. These models often provide more calibrated outputs on both in-distribution and out-of-distribution data by approximating epistemic and aleatoric uncertainty. Conformal prediction is a method that takes (possibly uncalibrated) predicted probabilities and produces prediction sets that follow attractive guarantees; namely marginal coverage for exchangeable data (Vovk et al., 2005). On out-of-distribution data, however, this guarantee no longer holds unless knowledge of the distribution shift is known *a priori* (Tibshirani et al., 2019; Barber et al., 2023). It is natural then to consider combining conformal prediction with Bayesian deep learning models to try and enjoy in-distribution guarantees and better calibration on out-of-distribution data. In fact, combination of the two methods has been used to correct for misspecifications in the Bayesian modeling process (Dewolf et al., 2023; Stanton et al., 2023), thereby improving coverage and thus trust in the broader machine learning system. However, we suggest that application of both methods in certain scenarios may be counterproductive and worsen performance on out-of-distribution examples (see Figure 1). To investigate this suggestion we evaluate the combination of Bayesian deep learning models and split conformal prediction methods on image classification tasks, where some of the test inputs are out-of-distribution. Our primary contributions are as follows:

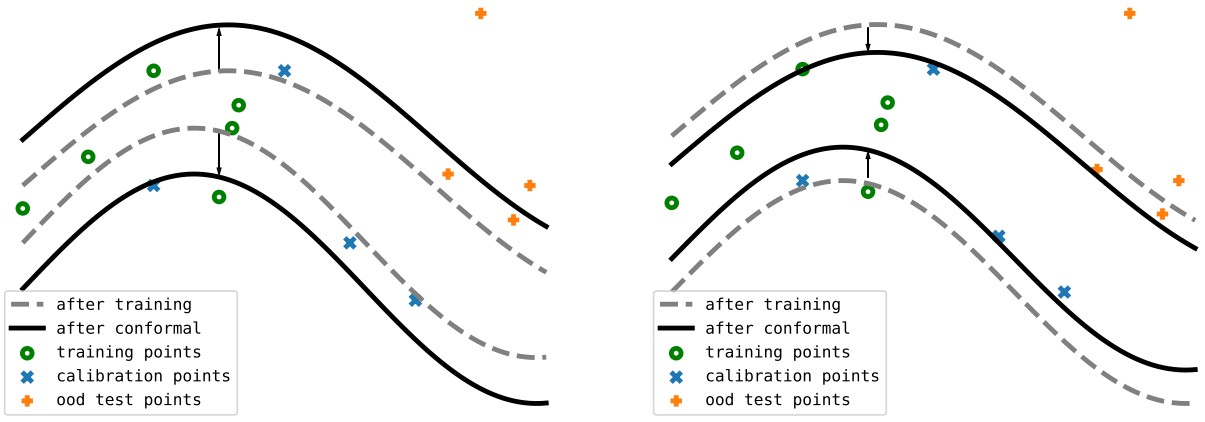

Figure 1: A conceptual illustration of how conformal prediction can help or harm out-of-distribution coverage for an error tolerance of 0.25. On the left is a conceptual illustration of how conformal prediction can make the overall machine learning system less confident after conformalizing a model that is overly confident on the calibration dataset. As a consequence, it gains coverage on out-of-distribution examples. The right conceptualizes the opposite direction and illustrates how conformal prediction can reduce coverage on out-of-distribution examples.

(i) Offer an explanation of how the under- or over-confidence of a model is tied to when conformal prediction may worsen or improve out-of distribution coverage.

(ii) Evidence to support the explanation in (i) and further demonstration of the *extent* to which conformal prediction affects out-of-distribution coverage in the form of two empirical evaluations focusing on the out-of-distribution coverage of Bayesian deep learning combined with split conformal prediction.

(iii) Practical recommendations for those using Bayesian deep learning and conformal prediction to increase the safety of their machine learning systems.

## 2 Preliminaries

### 2.1 Calibration

Modeling decisions factor into the *calibration* of the resultant predicted probabilities. Consider a vector of predicted probabilities produced by our model for input $\mathbf{x}$,

$$\hat{\boldsymbol{\pi}}(\mathbf{x}) = (\hat{\pi}_1(\mathbf{x}), ..., \hat{\pi}_K(\mathbf{x})), \;\; \sum_{k=1}^{K} \hat{\pi}_k(\mathbf{x}) = 1 \;\; \forall \mathbf{x} \in \boldsymbol{\mathcal{X}}$$

where $\boldsymbol{\mathcal{X}}$ is the sample space for $\mathbf{x}$ and $K$ is the number of possible labels $y$ can assume (i.e. the labels that can be assigned to $\mathbf{x}$). For a model to be *well-calibrated* (Zadrozny and Elkan, 2002; Vasilev and D'yakonov, 2023) the following must hold,

$$\mathbb{P}(y = i | \hat{\pi}_i(\mathbf{x}) = p) = p, \;\; \forall i \in \boldsymbol{\mathcal{Y}}, \forall p \in [0, 1],$$

where the probability is taken over the joint data distribution $p(\mathbf{x}, y)$ and $\boldsymbol{\mathcal{Y}}$ is the sample space for labels $y$. This says that on average over $p(\mathbf{x}, y)$, the predicted probability assigned to each class (not just the one assigned highest probability) represents the true probability of that class being the true label. If a model is well-calibrated, then we should be able to create prediction sets that achieve marginal coverage for an error tolerance $\alpha$ by creating a predictive credible set which we later abbreviate as *cred*. To do so, for each model output $\hat{\boldsymbol{\pi}}(\mathbf{x})$, we order the probabilities therein from greatest to least and continue adding the corresponding labels until the cumulative probability mass just exceeds $1 - \alpha$.

## 2.2 Bayesian Deep Learning

One important reason for miscalibration is the abstention of representing epistemic uncertainty (Wilson and Izmailov, 2020). In order to create more calibrated predicted probabilities for neural networks, a popular approach is Bayesian deep learning, where a major goal is to faithfully represent parameter uncertainty (a type of epistemic uncertainty). Parameter uncertainty[1] arises due to many different configurations of the model weights that can explain the training data, which happens especially in deep learning where we have highly expressive models (Wilson, 2020). Hence, Bayesian deep learning is an attractive approach to helping achieve calibration in practice, even on out-of-distribution examples. However, in order for the Bayesian approach to perform well a few important assumptions are required:

- Our observation model relating weights $\mathbf{w}$ and inputs $\mathbf{x}$ to labels $y$, $p(y|\mathbf{x}, \mathbf{w})$, is well-specified. This means that our observation model has the ability to produce the true data generating function.

- Our prior over weights $p(\mathbf{w})$ is well-specified. This means that when it is paired with the observation model, we induce a distribution over functions $p(f)$ that places sufficient probability on the true data generating function (MacKay, 2003).

- We often need to *approximate* the posterior distribution of the weights with respect to the training dataset, $p(\mathbf{w}|\mathcal{D})$, for many interesting observation models, including neural networks. It is required that this approximation to the true posterior is acceptable in the sense that both yield similar results for the task at hand. For example, in the predictive modeling scenario, the approximated and true posterior predictive should be nearly identical at the inputs that we will realistically encounter.

In this study we use a class of observation models (convolutional neural networks) and priors (zero-mean Gaussians) that have been shown to produce good inductive biases for image classification tasks (Wilson and Izmailov, 2020; Izmailov et al., 2021); and focus on varying the method of approximating the posterior over parameters.

## 2.3 Conformal Prediction

We restrict our attention to a subset of conformal prediction methods: *split* (or inductive) conformal prediction (Vovk, 2012). Split conformal prediction requires an extra held-out calibration dataset (which we denote $\mathcal{D}_{\text{cal}}$) to be used during a "calibration step". Importantly, split conformal prediction assumes exchangeability of the calibration and test set data. By allowing our final output to be a prediction set $\mathbb{Y} \subseteq (1, ..., K)$, conformal prediction guarantees the true class $y$ to be included on average with probability (confidence) $1 - \alpha$:

$$\Pr(y \in \mathbb{Y}) \geq 1 - \alpha, \tag{3}$$

where $\alpha$ is a user-chosen error tolerance and Pr reads "probability that". This guarantee is *marginal* in the sense that it is guaranteed on average with respect to the data distribution $p(\mathbf{x}, y)$ as well as the distribution over possible calibration datasets we could have selected. Not only does split conformal prediction guarantee the inequality in (3) but also upper bounds the coverage given that the scores $s_i$ have a continuous joint distribution. In particular, let $n_{\text{cal}}$ be the number of data pairs in the calibration dataset, then

$$\Pr(y \in \mathbb{Y}) \leq 1 - \alpha + \frac{1}{n_{\text{cal}} + 1}, \tag{4}$$

which is proved in Lei et al. (2018). Split conformal prediction methods work by first defining a *score* function that measures the disagreement between output probabilities $\hat{\boldsymbol{\pi}}(\mathbf{x})$ of a model and a label $y$. We denote a general score function as $s(\mathbf{x}, y)$, but it should be noted that the outputted scores depend on the underlying fitted model through the $\hat{\boldsymbol{\pi}}(\mathbf{x})$ it produces. The conformal method computes the scores on the held-out calibration dataset and then takes the

$$[(1 - \alpha)(1 + \frac{1}{|\mathcal{D}_{\text{cal}}|})]\text{-quantile}$$

---

[1] *Parameter* is a loaded term, and context is needed to precisely understand what it means. In this case, we mean the *weights* of the neural network; not the parameters that may govern the distribution over such weights.

of those scores which we denote $\tau$. Then, during test time, prediction sets are constructed by computing the score for each possible $y_i$, and including $y_i$ into the prediction set if its score is less than or equal to $\tau$. Informally, if the candidate label $y_i$ produces a score that conforms to what we have seen on the calibration dataset, it is included.

## 3  Motivation

We focus on the setting, as is often the case, where we expect to encounter out-of-distribution examples when we deploy our predictive systems in the real world. We are faced with the following binary choice: to either apply split conformal prediction to our model or to instead use non-conformal predictive credible sets. This choice will of course depend on our preferences. For instance, consider the scenario in which we want guaranteed marginal coverage on in-distribution inputs and additionally we would like to maximize the chance that our prediction sets cover the true labels for out-of-distribution inputs. What, then, are the benefits and risks associated with applying split conformal prediction? In beginning to answer this question, we provide an empirical evaluation alongside an explanation as to when applying split conformal prediction is expected to increase or decrease out-of-distribution coverage. The empirical evidence suggests that conformal prediction can significantly impact the out-of-distribution coverage one would otherwise see with predictive credible sets. And hence, understanding the interaction between certain predictive models and conformal prediction is important for the safe deployment of machine learning, especially in safety-critical scenarios.

While conformal prediction guarantees marginal coverage when the calibration data and test data are exchangeable, it loses its guarantees when encountering out-of-distribution data at test time[2]. Bayesian deep learning, on the other hand, has no such guarantees but has been shown to improve calibration (and thus coverage) on out-of-distribution inputs; a symptom of trying to quantify epistemic uncertainty. A natural desire, then, is to combine conformal prediction with Bayesian deep learning models in order to enjoy in-distribution guarantees while reaping the benefits of out-of-distribution calibration. To be clear, the main question this study intends to address is not how particular conformal prediction methods perform on out-of-distribution data. Instead, we address a more specific question: what are the benefits and risks associated with either applying split conformal prediction to our predictive model or passing up on using conformal prediction and simply using predictive credible sets when the predictive model is designed to be uncertainty-aware? In answering this question, we first discuss when, for a fixed predictive model, conformal prediction is expected to increase or decrease out-of-distribution coverage based on only on the calibration dataset. Afterward, we examine the *extent* to which the addition of conformal prediction increases or decreases the out-of-distribution coverage on real-world settings. As we will see, this depends on the behavior of the predictive uncertainty of the underlying model, particularly on out-of-distribution inputs.

The context is that we have a fixed predictive model and a split conformal prediction method. We would like to know whether this split conformal prediction method will increase (or decrease) marginal coverage on out-of-distribution data as compared to using credible sets. Conformal prediction does whatever it needs to guarantee the *desired coverage* on the calibration dataset; meaning it provides a lower bound (3) and an upper-bound (4) (see Figure 1). It does this by creating a threshold $\tau$ such that the routine "allow candidate label $y_i$ into the prediction set if the score $s(\mathbf{x}, y_i) \leq \tau$" produces sets that attain the desired marginal coverage on the calibration dataset. We say a predictive model is overconfident if its credible sets do not reach the desired coverage on the calibration dataset. The conformal method will then, by definition, create a $\tau$ such that *more* labels are allowed into the prediction sets in order to achieve the lower bound (3). And so on the calibration dataset, the conformal prediction sets will exhibit larger average set size than the credible sets. This will have the affect that, on any given future data, we can expect the average set size of the conformal sets to be larger than that of the credible sets. And hence, the conformal prediction sets will have a higher chance of achieving better marginal coverage. Conversely, we say the predictive model is underconfident if its credible sets exceed the desired coverage on the calibration dataset. The conformal method will then, by definition, create a $\tau$ such that *less* labels are allowed into the prediction sets in order to achieve the upper bound (4). And so on the calibration dataset, conformal prediction sets will exhibit smaller average set size than the credible sets. This will have the affect that, on any given future data, we can expect the average set

---

[2]This is true, unless, as mentioned before, there is *a priori* knowledge about the out-of-distribution data (Tibshirani et al., 2019; Barber et al., 2023)

size of credible sets to be larger than conformal prediction sets. And thus, credible sets will have a higher chance of achieving better marginal coverage.

It follows from this discussion that, for a fixed model, the prediction set method we expect to have the greatest out-of-distribution coverage will have largest average set size on the calibration dataset. Put another way, increasing the chance of out-of-distribution coverage without any knowledge of the test distribution will be at the expense of larger average set size on in-distribution data. We have just proposed an explanation as to when conformal prediction will likely increase (or decrease) marginal coverage on out-of-distribution data in the context of a fixed predictive model. In Section 6 we empirically examine the *extent* to which conformal prediction can either help or harm out-of-distribution coverage as compared to credible sets. Furthermore, we examine this extent across many different predictive models that exhibit different behavior with respect to predictive uncertainty on out-of-distribution data.

## 4 Related Work

The analysis of combining more traditional Bayesian models with conformal prediction has been studied in Wasserman (2011) and Hoff (2021). Combining *full* (not *split*) Bayesian *deep learning* with conformal *regression* has been studied in the context of efficient computation of conformal Bayesian sets and Bayesian optimization (Fong and Holmes, 2021; Stanton et al., 2023). Both of these works consider settings with non-exchangeable data but assume *a priori* knowledge on the type of distribution shift.

Theoretical foundations of using conformal prediction with non-exchangeable data can admit very powerful guarantees but have thus far assumed *a priori* knowledge about the distribution shift (Tibshirani et al., 2019; Angelopoulos et al., 2022; Fannjiang et al., 2022; Barber et al., 2023). Additionally, other powerful conformal algorithms have been posed to deal with out-of-distribution examples but require an additional model to detect those out-of-distribution examples (Angelopoulos et al., 2021). Dewolf et al. (2023) evaluate conformal prediction combined with Bayesian deep learning models but for regression tasks and with relatively low-dimensional inputs (no greater than 280 features). Additionally, Kompa et al. (2021) look at the coverage of prediction sets from various Bayesian deep learning methods and mention conformal prediction but do not include conformal prediction in their empirical analysis. We are not aware of a study examining the interaction between conformal prediction and Bayesian deep learning methods as it relates to the coverage of unknown, out-of-distribution examples for a task that deep learning models have excelled—image classification. We not only evaluate the combination of Bayesian deep learning and conformal prediction on image classification but also provide an intuitive explanation as to why, in certain scenarios, conformal prediction can actually *harm* out-of-distribution coverage we would otherwise see with non-conformal predictive sets.

## 5 Evaluation & Method Details

### 5.1 Predictive Modeling Methods

Consider a probabilistic conditional model $p(y, \mathbf{w}|\mathbf{x}) = p(y|\mathbf{x}, \mathbf{w})p(\mathbf{w})$ where $\mathbf{w}$ are the weights, $p(\mathbf{w})$ is a prior distribution, and $p(y|\mathbf{x}, \mathbf{w})$ is the probability of $y$ given our weights $\mathbf{w}$ and fixed inputs $\mathbf{x}$. Denote the training dataset as $\mathcal{D}$. Then the posterior predictive distribution can be written as

$$p(y|\mathbf{x}, \mathcal{D}) = \int p(y|\mathbf{x}, \mathbf{w})p(\mathbf{w}|\mathcal{D})d\mathbf{w}. \tag{5}$$

We implement three methods for approximating (5), which is sometimes referred to as the *Bayesian model average* (Wilson and Izmailov, 2020).

**Stochastic Gradient Descent** (SGD) typically involves finding the maximum a posteriori (MAP) estimate for $\mathbf{w}$. In the context of (1), this means we approximate $p(\mathbf{w}|\mathcal{D}) \approx \delta(\mathbf{w} - \hat{\mathbf{w}}_{\text{MAP}})$ where

$$\hat{\mathbf{w}}_{\text{MAP}} = \underset{\mathbf{w}}{\text{argmax}} \ \{\log p(\mathbf{w}|\mathcal{D})\} = \underset{\mathbf{w}}{\text{argmax}} \ \{(\log p(\mathcal{D}|\mathbf{w}) + \log p(\mathbf{w}) + \text{constant})\}.$$

Neural networks trained via stochastic gradient descent have been found to often be uncalibrated by being overly confident in their predictions, especially on out-of-distribution examples (Guo et al., 2017).

**Deep ensembles** (ENS) works by combining the outputs of multiple neural networks with different initializations (Lakshminarayanan et al., 2017). The idea is that the variation in their respective outputs can represent epistemic uncertainty. In this case, we implicitly approximate the posterior and simply average the hypotheses generated by each model in the ensemble:

$$p(y|\mathbf{x}, \mathcal{D}) \approx \frac{1}{J} \sum_{j=1}^{J} p(y|\mathbf{x}, \hat{\mathbf{w}}_{\mathrm{MAP}_j})$$

where $\hat{\mathbf{w}}_{\mathrm{MAP}_j}$ is the stochastic gradient solution for model $j$ in the ensemble of $J$ models. Deep ensembles can be viewed as an approximation to the Bayesian model average (5) (Wilson and Izmailov, 2020).

**Mean-field Variational Inference** (MFV) seeks to approximate the posterior with a variational distribution (Blei et al., 2017):

$$p(\mathbf{w}|\mathcal{D}) \approx q(\mathbf{w}|\hat{\boldsymbol{\theta}}) = \prod_{i=1}^{m} q(w_i|\hat{\theta}_i) \tag{6}$$

which comes from the mean-field assumption, and $q$ is usually a simple distribution (e.g. a Gaussian) [3]. To make the approximation in (6), we maximize with respect to $\boldsymbol{\theta}$ a function equivalent to the KL divergence between $p(\mathbf{w}|\mathcal{D})$ and $q(\mathbf{w}|\boldsymbol{\theta})$ up to a constant. It is usually termed the *evidence lower bound* (ELBO). We can write the objective as

$$\hat{\boldsymbol{\theta}} = \operatorname*{argmax}_{\boldsymbol{\theta}} \{\mathrm{ELBO}(\boldsymbol{\theta}, \mathcal{D})\} = \operatorname*{argmax}_{\boldsymbol{\theta}} \left\{ \left( \underbrace{\mathbb{E}_{\mathbf{q}(\mathbf{w}|\boldsymbol{\theta})}[\log p(\mathcal{D}|\mathbf{w})]}_{\text{Expected log likelihood of data}} + \overbrace{\mathrm{KL}[q(\mathbf{w}|\boldsymbol{\theta}) \,||\, p(\mathbf{w})]}^{\text{KL between variational posterior and prior}} \right) \right\}.$$

The ELBO objective is a tradeoff between two terms. The expected log likelihood of the data prefers $q(\cdot)$ place its mass on the maximum likelihood estimate while the KL divergence term prefers $q(\cdot)$ stay close to the prior (Blei et al., 2017). Bayesian neural networks for classification have been shown to be generally underconfident by overestimating aleatoric uncertainty (Kapoor et al., 2022). Indeed, we find this is the case for both our experiments (see Figure 5).

**Stochastic Gradient Hamiltonian Monte Carlo** (SGMHC) is a Markov Chain Monte Carlo (MCMC) method. In the context of inferring the posterior over parameters $\mathbf{w}$, the basic premise of MCMC is to construct a Markov chain on the parameter space $\mathcal{W}$ whose stationary distribution is the posterior of interest $p(\mathbf{w}|\mathcal{D})$. Hamiltonian Monte Carlo (HMC) is a popular MCMC technique that uses gradient information and auxiliary variables to better perform in high-dimensional spaces (see 12.4 and 12.5 of Murphy (2023)). SGHMC attempts to unify the efficient exploration of HMC and the computational feasibility of *stochastic* gradients (Chen et al., 2014). To this end, it follows stochastic gradient HMC sampling with an added friction term that counters the effects of the (noisy) stochastic gradients. This results in a second-order Langevin dynamical system, which is closely related to stochastic gradient Langevin dynamics (SGLD) (Welling and Teh, 2011). SGLD uses first-order dynamics, and can be viewed as a limiting case of the second-order dynamics of SGHMC.

**Laplace Approximation** (LAPLACE) provides a Gaussian approximation to $p(\theta|\mathcal{D})$ by casting the posterior in terms of an energy function $E(\mathbf{w}) = -\log p(\mathbf{w}, \mathcal{D})$ and then Taylor approximating this energy function around the MAP solution $\mathbf{w}_{\mathrm{MAP}}$. The result is a multivariate Gaussian approximation of the posterior with its mean as $\mathbf{w}_{\mathrm{MAP}}$ and its covariance matrix as the inverse Hessian of the energy function $E$ taken with respect to $\mathbf{w}$ and evaluated at $\mathbf{w}_{\mathrm{MAP}}$. That is, we approximate the posterior as

$$p(\mathbf{w}|\mathcal{D}) \approx \mathcal{N}(\mathbf{w}|\mathbf{w}_{\mathrm{MAP}}, \mathbf{H}^{-1}).$$

See 7.4.3 in Murphy (2023) for a derivation of the Laplace approximation. There are many different variants of the Laplace approximation for deep learning. These variants usually stem from how one approximates

---

[3]Note that we are considering only *global* variable models, not *local*. Local variables, often denoted as $z_i$, each only affect a corresponding data point. As such, the number of $z_i$ scales linearly with the size of the dataset. Meanwhile, global variables affect each data point in the same way.

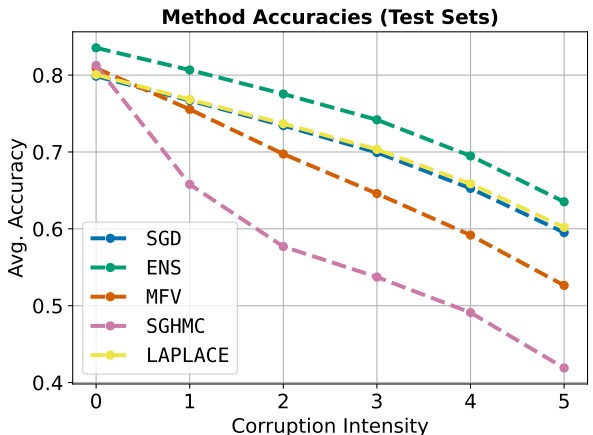

| Credible Set Coverage On The Calibration Dataset | | | | | |
|---|---|---|---|---|---|
| | SGD | ENS | MFV | SGHMC | LAPLACE |
| **0.05** Error | 88% | 95% | 99% | 99% | 99% |
| **0.01** Error | 92% | 97% | 100% | 100% | 100% |

| Average Set Size On The Calibration Dataset | | | | | |
|---|---|---|---|---|---|
| **0.05** Error | SGD | ENS | MFV | SGHMC | LAPLACE |
| *cred* | 1.38 | 1.76 | **2.97** | **2.82** | **4.96** |
| *thr* | 2.12 | 1.72 | 1.85 | 1.80 | 2.06 |
| *aps* | **2.19** | **1.92** | 2.37 | 2.25 | 2.66 |
| **0.01** Error | SGD | ENS | MFV | SGHMC | LAPLACE |
| *cred* | 1.69 | 2.20 | **4.33** | **4.13** | **8.56** |
| *thr* | 4.10 | **3.32** | 3.30 | 3.02 | 4.41 |
| *aps* | **4.12** | 3.25 | 3.58 | 3.44 | 4.96 |

(a) **CIFAR10** *accuracy* plot. The accuracy plot shows, for each corruption intensity, the average accuracy over all corrupted datasets at that intensity.

(b) **CIFAR10**: The first table displays the credible set coverage on the calibration dataset to indicate over- and under-confident predictive methods. The second table displays the average set sizes on the calibration dataset. For each predictive model method, the prediction set method with highest average set size is bolded.

Figure 2: The accuracy plot and calibration dataset results for the CIFAR10 experiment.

**H** due to its infeasible computational requirements for large models as well as the form of the posterior predictive distribution (Daxberger et al., 2021). A popular variant is determined by only considering the last layer weights, Kronecker factorizing the Hessian (Ritter et al., 2018), and using a linearized predictive distribution (Immer et al., 2021). We implement this Laplace variant for our experiments in Section 6.1.

## 5.2 Conformal Prediction Methods

We implement two common split conformal prediction methods.

**Threshold prediction sets** (*thr*) uses the following score function (Sadinle et al., 2019):

$$s(\mathbf{x}, y) = 1 - \hat{\pi}_y(\mathbf{x}).$$

That is, the score for an input $\mathbf{x}$ with true label $y$ is one minus the probability mass the model assigns to the true label $y$. This procedure only takes into account the probability mass assigned to the *correct label*.

**Adaptive prediction sets** (*aps*) uses the following score function (Romano et al., 2020):

$$s(\mathbf{x}, y) = \hat{\pi}_1(x) + \cdots + U\hat{\pi}_y(x),$$

where $\hat{\pi}_1(x) \geq \cdots \geq \hat{\pi}_y(x)$ and $U$ is a uniform random variable in $[0, 1]$ to break ties. That is, we order the probabilities in $\hat{\boldsymbol{\pi}}(\mathbf{x})$ from greatest to least and continue adding the probabilities, stopping after we reach the probability associated with the correct label $y$.

## 5.3 Evaluation Measures

A prediction set is any subset of possible labels. We would like a collection of prediction sets to have certain minimal properties. We would first like the collection to be *marginally covered*: given a user-specified error tolerance $\alpha$, the average probability that the true class $y$ is contained in the given prediction sets is $1 - \alpha$. We would also like each prediction set in the collection to be *small*: the prediction sets should contain as few labels as possible without losing coverage. Hence, we evaluate on the basis of the marginal coverage and average size of a collection of prediction sets emitted by applying split conformal prediction to Bayesian deep learning model outputs.

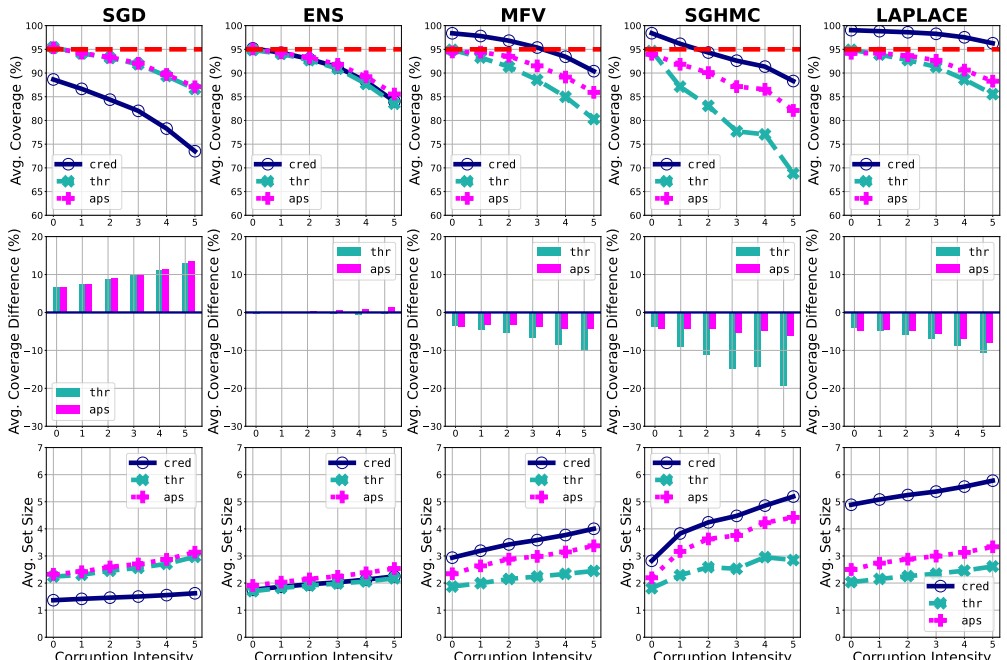

Figure 3: CIFAR10 and CIFAR10-Corrupted results at the **0.05** error tolerance. **Row 1** and **Row 3** illustrate the average coverage and average set size (respectively) of each prediction set method (*thr*, *ap*, simple predictive (*cred*)) for each predictive modeling method. To explicitly indicate the extent to which conformal prediction effects simple predictive sets, **Row 1** shows the average coverage difference between conformal prediction methods (*thr*, *ap*) and simple predictive credible sets (*cred*).

**Measuring marginal coverage**: for a given test set $\mathcal{D}_{\text{test}} = \{(\mathbf{x}_i, y_i)\}_{i=1}^{n_{\text{test}}}$ and prediction set function $\mathbb{Y}(\mathbf{x})$ that maps inputs to a prediction set, we measure marginal coverage as

$$\frac{1}{n_{\text{test}}} \sum_{i=1}^{n_{\text{test}}} \mathbf{1}\left\{y_i \in \mathbb{Y}(\mathbf{x}_i)\right\},$$

where $\mathbf{1}$ denotes the indicator function. This is just the fraction (percent) of prediction sets that covered the corresponding true label $y_i$.

**Measuring set size**: for a collection of prediction sets $\{\mathbb{Y}(\mathbf{x}_i)\}_{i=1}^{n_{\text{test}}}$, we measure the set size as simply the average size of the prediction sets:

$$\frac{1}{n_{\text{test}}} \sum_{i=1}^{n_{\text{test}}} |\mathbb{Y}(\mathbf{x}_i)|.$$

## 6 Experiments

### 6.1 CIFAR10-Corrupted

Our first experiment loosely follows the setup of Izmailov et al. (2021) and Ovadia et al. (2019). We train an AlexNet inspired model on CIFAR10 by means of stochastic gradient descent, deep ensembles, and mean-field variational inference (Krizhevsky et al., 2009; 2012). We then take 1000 examples (without replacement) from the CIFAR10 test set for a calibration dataset. We use this calibration dataset to determine thresholds $\tau$ that we use for conformal methods *thr* and *aps*, respectively. We then evaluate the marginal coverage and

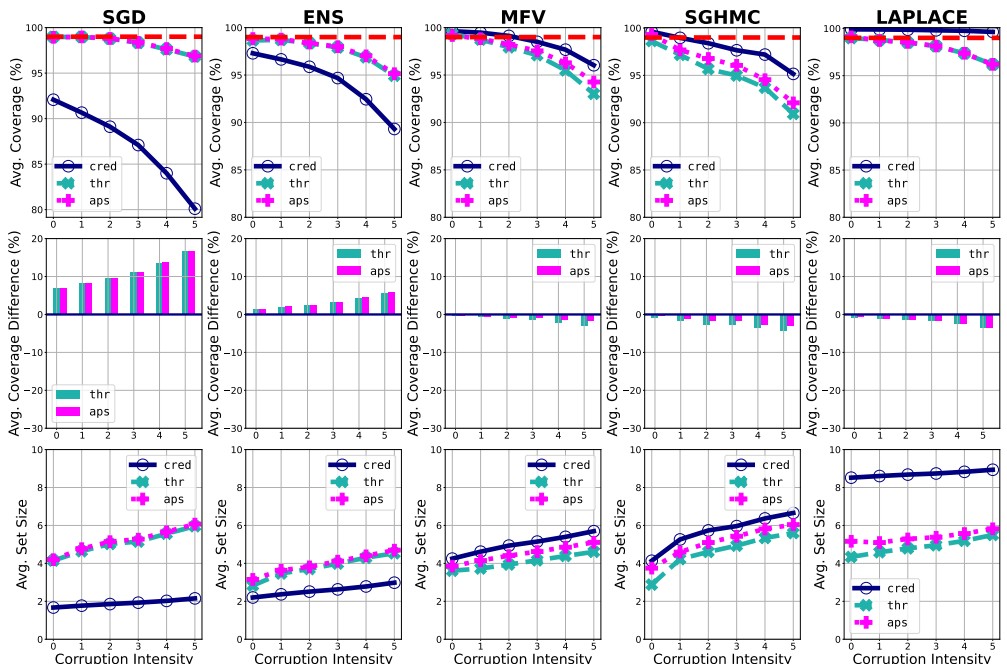

Figure 4: **CIFAR10**: CIFAR10 and CIFAR10-Corrupted results at the **0.01** error tolerance. **Row 1** and **Row 3** illustrate the average coverage and average set size (respectively) of each prediction set method (*thr*, *ap*, simple predictive) for each predictive modeling method. To explicitly indicate the extent to which conformal prediction effects simple predictive sets, **Row 2** shows the average coverage difference between conformal prediction methods (*thr*, *ap*) and simple predictive credible sets.

average size of the predictive credible sets (*cred*), *thr* sets, and *aps* sets that were produced for every single CIFAR10-Corruped dataset (Hendrycks and Dietterich, 2019) at every intensity[4].

We first look at the credible set coverage of each predictive model method on the *calibration dataset*, which is shown in Figure 2b. Recall from Section 3 that we say a predictive model is overconfident if the coverage of its credible sets on the calibration dataset are less than the desired coverage, and underconfident if they exceed the desired coverage. In Figure 2b we see that SGD is overconfident at both error tolerances, ENS is overconfident at the 0.01 error tolerance and neither overconfident or underconfident at the 0.05 error tolerance, and MFV, SGHMC, and LAPLACE are all underconfident. We expect then that conformal prediction set methods will increase out-of-distribution coverage for SGD (and ENS at the 0.01 error tolerance), while decrease out-of-distribution coverage for MFV, SGHMC, and LAPLACE. However, to what extent is this the case? To analyze this empirically, we show the average[5] marginal coverage, the average *difference* in marginal coverage between conformal prediction methods and simple predictive credible sets, and average set size across datasets at each intensity in Figures 3 and 4. The average accuracies for each predictive modeling method are shown in Figure 2a.

**Remarks:**

- In the context of the methods evaluated, if a top priority is to capture true labels *even* on unknown out-of-distribution inputs, one is generally better off using "uncertainty-aware" methods such as MFV, SGHMC, and LAPLACE *without* conformal prediction. However, these three methods vary widely in how large their average set size is in order to achieve good out-of-distribution coverage.

---

[4]We take out those semantically similar images from the corrupted test set that we used for the calibration dataset to ensure no data leakage.

[5]We take the average across different calibration and test set splits as well as different datasets that comprise the CIFAR10-Corrupted dataset. To see both specific results on each of the different datasets (types of corruptions) and plots with error bars, see the Appendix.

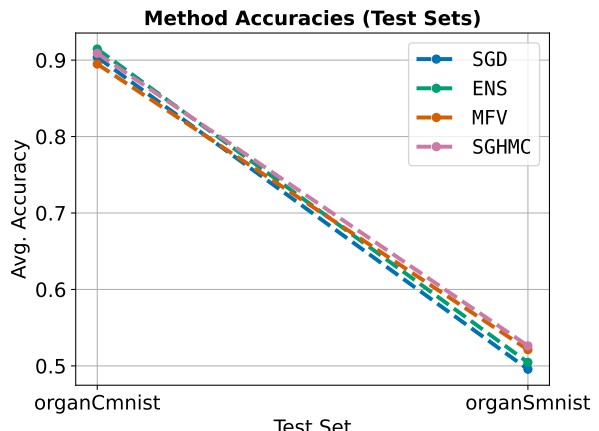

| Credible Set Coverage On The Calibration Dataset | | | | |
|---|---|---|---|---|
| | SGD | ENS | MFV | SGHMC |
| **0.05** Error | 94% | 96% | 99% | 99% |
| **0.01** Error | 97% | 98% | 100% | 100% |

| Average Set Sizes On The Calibration Dataset | | | | |
|---|---|---|---|---|
| **0.05** Error | SGD | ENS | MFV | SGHMC |
| *cred* | 1.20 | 1.34 | **2.97** | **1.87** |
| *thr* | 1.33 | 1.22 | 1.18 | 1.15 |
| *aps* | **1.59** | **1.58** | 1.95 | 1.57 |
| **0.01** Error | SGD | ENS | MFV | SGHMC |
| *cred* | 1.65 | 1.94 | **5.68** | **2.96** |
| *thr* | 3.46 | 3.02 | 2.94 | 1.88 |
| *aps* | **4.96** | **5.32** | 4.85 | 2.87 |

(a) **MedMNIST** *accuracy* plot. It is evaluated for the in-distribution organ**C**mnist dataset and the out-of-distribution organ**S**mnist dataset.

(b) **MedMNIST**: The first table displays the credible set coverage on the calibration dataset to indicate over- and under-confident predictive methods. The second table displays the average set sizes on the calibration dataset. For each predictive model method, the prediction set method with highest average set size is bolded.

Figure 5: The accuracy plot and calibration dataset results for the MedMNIST experiment.

- For a fixed predictive model, the prediction set method with greatest out-of-distribution coverage is that with the largest average set size on those out-of-distribution inputs as well as on the calibration dataset (see Figure 2b for average set size on the calibration dataset).

- Across all predictive models, larger average set size of a prediction set method paired with a predictive model will usually mean better out-of-distribution coverage, but there can be exceptions arising from how different predictive models exhibit uncertainty on out-of-distribution inputs. For example, MFV credible sets are smaller on out-of-distribution inputs than SGHMC credible sets, but MFV credible sets achieve better coverage.

- A small change in set size on in-distribution data can have drastic effects on the out-of-distribution coverage, especially on inputs that are far from the training distribution. For example (see Figure 3) an $\approx 1$ decrease in average set size due to performing *thr* conformal prediction results in an $\approx 20\%$ drop in out-of-distribution coverage as compared to SGHMC credible sets and an $\approx 10\%$ drop in out-of-distribution coverage as compared MFV credible sets.

- Being more robust to the corruptions in terms of *accuracy* (see Figure 2a) does not immediately translate to better coverage on out-of-distribution examples (see Figures 3 and 4). For instance, both MFV and SGHMC perform the worst in terms of accuracy on the out-of-distribution test sets, however they perform better than SGD and ENS in terms of out-of-distribution coverage. And importantly, this is not due to being trivially underconfident, i.e. producing sets of all possible labels for each input.

- A majority of the time, *thr* and *aps* behave similarly in terms of average set size and out-of-distribution coverage. However, there are notable exceptions where *thr* produces smaller sets on out-of-distribution inputs which in turn more negatively impacts out-of-distribution coverage (see, for example, Figure 3, SGHMC and MFV).

We do not argue that any method (or combination thereof) is always preferred over another. Indeed, what one decides upon will depend on many factors. Two important factors (that this study highlights) is both the frequency of out-of-distribution data encountered by a deployed model as well as *how* far these data may be from the training distribution. It is straightforward why the former is important, and the latter is important

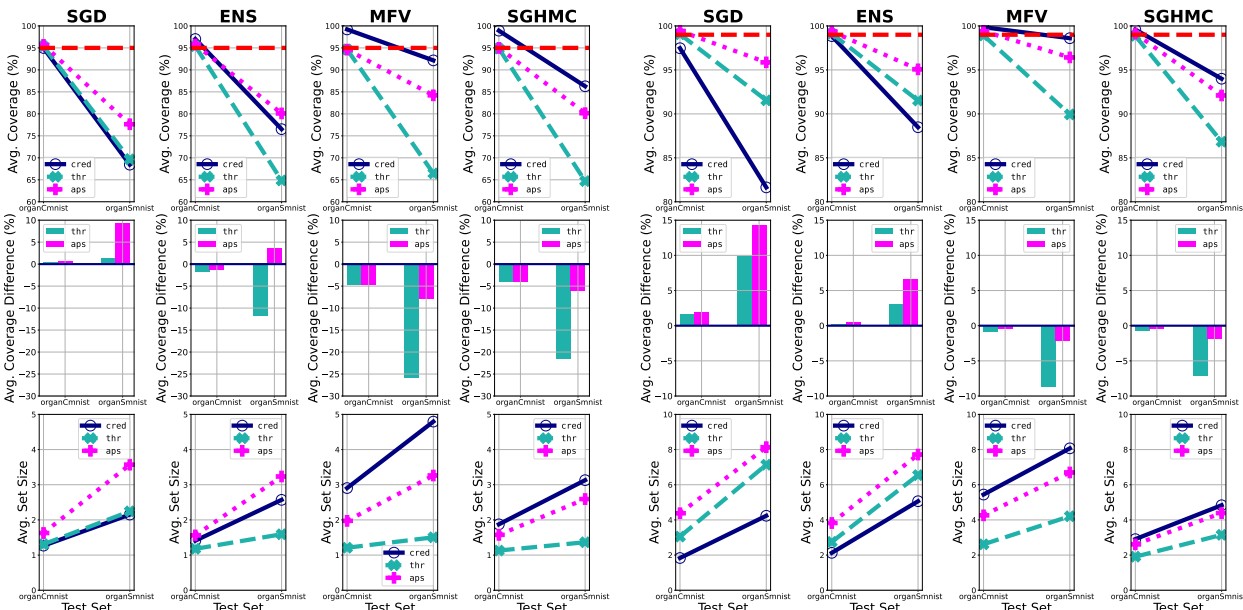

(a) Error tolerance of 0.05 (i.e. 0.95 desired coverage) which is denoted by the dashed horizontal red line.

(b) Error tolerance of 0.01 (i.e. 0.99 desired coverage) which is denoted by the dashed horizontal red line.

Figure 6: **MedMNIST**: Average marginal coverage and average set size for the in-distribution organ**C**mnist dataset (denoted *in* on the plot) and for the out-of-distribution organ**S**mnist dataset (denoted *out* on the plot.)

because we see a large disparity between the type of coverage we would see with conformal prediction as opposed to without it on data that is far from the training distribution (e.g. see Figure 3, SGHMC). We also illustrate that the error tolerance we choose does not represent our *true* desires if we believe out-of-distribution data will be encountered. While this is to be expected since our conformal prediction methods contain no guarantees for out-of-distribution data, we demonstrate the *extent* to which this error tolerance is violated, and how in some cases it can be mitigated by foregoing the use of conformal prediction.

## 6.2 MedMNIST

Our second experiment provides a more realistic safety-critical scenario: 11-class classification of radiology scans. We train a ResNet18 model on the organ**C**mnist dataset of MedMNIST with the same three modeling methods (He et al., 2016; Yang et al., 2023). Conformal calibration is done using the first 500 examples from the test set of organ**C**mnist. We then evaluate the coverage and size on the remaining examples from the test set of organ**C**mnist for all three prediction set methods. The organ**S**mnist dataset has the same classes but is the result of different *views* of the subject of interest in the radiology scan. Thus, it is from a different distribution, and so we also evaluate the coverage and size on the test set of organ**S**mnist to see how the combination of conformal methods and Bayesian deep learning affects out-of-distribution coverage.

As in the previous experiment, we first note the credible set coverage of each predictive model method on the *calibration dataset* (shown in Figure 5b). Similar to the CIFAR10 experiment, SGD is overconfident and MFV and SGHMC are underconfident. ENS is underconfident at the 0.05 error tolerance but overconfident at the 0.01 error tolerance. We expect then that, in terms of out-of-distribution coverage, conformal prediction set methods will most likely *help* SGD, most likely *harm* MFV and SGHMC, and most likely *harm* ENS at the 0.05 error tolerance but *help* ENS at the 0.01 error tolerance. And again we ask, to what extent is this the case? In answering this, the average marginal coverage and average set size for each dataset is shown in Figure 6a for the 0.05 error tolerance and Figure 6b for the 0.01 error tolerance. The accuracies of the deep learning methods are shown in Figure 5a. The *in*-distribution dataset is organ**C**mnist and the *out*-of-distribution dataset is organ**S**mnist.

**Remarks:**

- In the context of the methods evaluated, if a top priority is to capture true labels *even* on unknown out-of-distribution inputs, one is generally better off using "uncertainty-aware" methods such as MFV and SGHMC *without* conformal prediction. An exception is at the 0.01 error tolerance where ENS *aps* sets and SGD *aps* sets attain slightly better coverage than SGHMC, although at the cost of significantly larger average set size.

- For a fixed predictive model, the prediction set method with greatest out-of-distribution coverage is that with the largest average set size on those out-of-distribution inputs as well as on the calibration dataset.

- Across all predictive models, larger average set size of a prediction set method paired with a predictive model will usually mean better out-of-distribution coverage, but there can be exceptions arising from how different predictive models exhibit uncertainty on out-of-distribution inputs. For example at the 0.01 error tolerance, MFV *aps* sets are smaller on out-of-distribution inputs than ENS *aps* sets but achieve better coverage.

- A small change in average set size on in-distribution data can have drastic effects on the out-of-distribution coverage. For example, an $\approx 1$ decrease in average set size due to *thr* with SGHMC results in $\approx 21\%$ drop in out-of-distribution coverage (Figure 6a). On the flip side, an $\approx .3$ increase in average set size due to *aps* with SGD results in $\approx 10\%$ increase in out-of-distribution coverage (Figure 6a).

## 7 Discussion

**Limitations and Future Work:** We recognize that we evaluated and compared only a few of the many conformal and Bayesian methods. Furthermore, although the results presented here add an important dimension to the practical considerations of combining conformal and Bayesian deep learning methods, there are many other questions that remain to be answered (e.g. adaptivity gains from using Bayesian deep learning with conformal prediction). We developed experiments that provide evidence for the explanation offered in Section 3 (see also Figure 1) which consequently demonstrated that certain modeling and data scenarios can seriously impact the benefit of conformal prediction. Future work may include developing diagnostics or practical checks that suggest one is in a particular scenario in which the utility of conformal prediction can be predicted. Future work might also include further evaluations with additional measures such as size-stratified coverage (Angelopoulos et al., 2020), and further mathematical analysis. Such analysis might provide additional insights into when and how to combine certain conformal prediction and Bayesian deep learning methods.

**Conclusion:** We demonstrated important scenarios in which conformal prediction can decrease the out-of-distribution coverage one would otherwise see with simple predictive credible sets. Importantly, we also demonstrate the *extent* to which conformal prediction does so. We also saw that in some cases it is not realistic to think that the error tolerance we select will be honored. We hope that this study motivates the need for better evaluation strategies for Bayesian deep learning models. Echoing some of the arguments made in Kompa et al. (2021), frequentist coverage of both in-distribution and out-of- distribution examples for Bayesian deep learning models provides a nuanced and practical representation of both the calibration of the models and the benefits of using conformal prediction in realistic settings. These are all of immediate practical importance for a wide range of application areas, particularly in those where unsafe mistakes can incur a large cost. If strong guarantees of coverage are desired, then one may consider Bayesian deep learning, conformal prediction, or both, in an effort to provide those guarantees. Knowledge of the scope of application, an assessment to identify breaking important assumptions (e.g. out-of-distribution data), and expected use may help decide the methods that should be applied. Being aware of these results and using the conclusions will better equip engineers in creating safer machine learning systems.

**Disclosure of Funding:** This research was funded by MITRE's Independent Research and Development Program

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

# APPENDIX

## A   TRAINING DETAILS

For both experiments we train SGD and MFV for 5 different seeds. ENS is the result of combining 5 stochastic gradient descent model states from the different 5 seeds. Due to the computational cost of running SGHMC, we only trained it for a single seed. We measure the validation accuracy (or in the case of SGHMC collect a sample) every 10 epochs. For the CIFAR10 experiment we run SGHMC for 1000 burnin epochs and then 9000 epochs therafter resulting in 900 total samples. For the MedMNIST experiment we run SGHMC for 1000 burnin epochs and then 10000 epochs thereafter resulting in 1000 total samples.

For evaluation of SGD and MFV we select the model state that attains the best validation accuracy. If the best validation accuracy is shared between multiple checkpoints, we use the model state from the earliest checkpoint amongst those that are tied.

LAPLACE is a result starting with one of our pretrained SGD solutions and then conducting the laplace approximation with the package from Daxberger et al. (2021). In particular, we compute a laplace approximation for the last-layer with the *kron* approximation to the Hessian. Furthermore, we use the *glm* alongside the *probit* approximation for the posterior predictive. See Daxberger et al. (2021) for more details. We only use the LAPLACE method in the CIFAR10 experiment.

### A.1   CIFAR10

**Dataset:**   The CIFAR10 dataset contains 60,000 $32 \times 32 \times 3$ RGB images in 10 classes, where each class contains 6,000 images each. There are 50,000 training images (5,000 images per class) and 10,000 test images (1,000 images per class). We take 5% of the original training dataset ($0.05 \times 50,000 = 2,500$) examples as a validation set, and leave the remaining 95% ($50,000 - 2,500 = 47,500$) examples for training. For preprocessing, we normalize the images with mean $(0.49, 0.48, 0.44)$ and standard deviation $(0.2, 0.2, 0.2)$ for each of the 3 channels. This is taken from the code repository of (Izmailov et al., 2021). We performed *no* data augmentation.

**Base Model:**   We use an AlexNet inspired convolutional neural network as a base model, which is taken from the code repository of (Izmailov et al., 2021).

**Training Hyperparameters:**   The following tables illustrate the training hyperparameters for the CIFAR10 experiment.

Table 1: Hyperparameters for SGD and MFV. The initial $\sigma$ is the initial value of the standard deviation of the per-parameter Gaussians for mean-field variational inference.

(a) SGD Training Hyperparameters

| Name | Value |
|------|-------|
| seeds | $\{1, \ldots, 5\}$ |
| batch size | 80 |
| epochs | 100 |
| weight decay | 5.0 |
| temperature | 1.0 |
| learning rate schedule | cosine |
| checkpoint frequency | 10 |
| initial step size | 8e-7 |
| optimizer | sgd |
| momentum decay | 0.9 |

(b) MFV Training Hyperparameters.

| Name | Value |
|------|-------|
| seeds | $\{1, \ldots, 5\}$ |
| batch size | 80 |
| epochs | 100 |
| weight decay | 5.0 |
| temperature | 1.0 |
| learning rate schedule | cosine |
| checkpoint frequency | 10 |
| initial step size | 4e-4 |
| optimizer | Adam |
| initial $\sigma$ | 0.01 |
| # samples | 1 |

Table 2: Hyperparameters for SGHMC

| Name | Value |
|---|---|
| seeds | 1 |
| batch size | 80 |
| epochs | 10000 |
| weight decay | 5.0 |
| temperature | 1.0 |
| learning rate schedule | cyclical |
| cycle epochs | 75 |
| checkpoint frequency | 10 |
| initial step size | 3e-6 |
| momentum decay | 0.9 |
| preconditioner | RMSprop |

## A.2 MedMNIST

**Dataset:** MedMNIST contains many standardized datasets of biomedical images (Yang et al., 2023). We train on one of these datasets: organ**C**mnist. This dataset is part of a larger cohort of three datasets which are based on the 3D CT images from the Liver Tumor Segmentation Benchmark (Bilic et al., 2023). The larger cohort is {organ**A**mnist, organ**C**mnist, organ**S**mnist }, where **A**,**C**, and **S** are short for Axial, Coronal, and Sagittal. These describe different *views* of the CT scan (see Figure 7). We use the pre-specified training and validation sets provided by MedMNIST. These contain 13,000 training examples and 2,392 validation examples. Each image is grayscale. For preprocessing, we normalize the images with mean 0.49 and standard deviation 0.2 for the single channel. We performed *no* data augmentation.

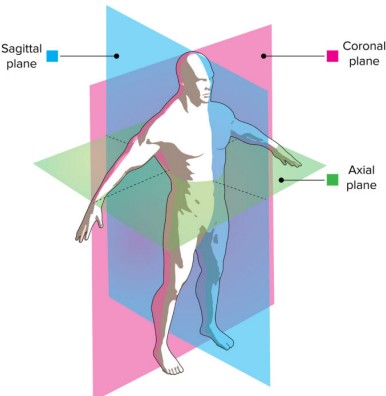

Figure 7: An illustration describing the axial, coronal, and sagittal views. `https://anatomytool.org/content/lecturio-drawing-sagittal-coronal-and-transverse-plane-english-labels`

**Base Model:** We use a ResNet18 neural network as a base model (He et al., 2016), which is a built-in model in the Haiku library (Hennigan et al., 2020).

**Training Hyperparameters:** The following tables illustrate the training hyperparameters for the MedMNIST experiment.

Table 3: Hyperparameters for SGD and MFV. The initial $\sigma$ is the initial value of the standard deviation of the per-parameter Gaussians for mean-field variational inference.

(a) SGD Training Hyperparameters

| Name | Value |
|---|---|
| seeds | $\{1, \ldots, 5\}$ |
| batch size | 80 |
| epochs | 100 |
| weight decay | 10.0 |
| temperature | 1.0 |
| learning rate schedule | cosine |
| checkpoint frequency | 10 |
| initial step size | 6e-6 |
| optimizer | sgd |
| momentum decay | 0.9 |

(b) MFV Training Hyperparameters.

| Name | Value |
|---|---|
| seeds | $\{1, \ldots, 5\}$ |
| batch size | 80 |
| epochs | 100 |
| weight decay | 10.0 |
| temperature | 1.0 |
| learning rate schedule | cosine |
| checkpoint frequency | 10 |
| initial step size | 1e-4 |
| optimizer | Adam |
| initial $\sigma$ | 0.01 |
| # samples | 1 |

Table 4: Hyperparameters for SGHMC

| Name | Value |
|---|---|
| seeds | 1 |
| batch size | 80 |
| epochs | 11000 |
| weight decay | 5.0 |
| temperature | 1.0 |
| learning rate schedule | cyclical |
| cycle epochs | 75 |
| checkpoint frequency | 10 |
| initial step size | 1e-5 |
| momentum decay | 0.9 |
| preconditioner | RMSprop |

## B  EVALUATION DETAILS

We first note that we use 30 samples to approximate the posterior predictive density when using mean-field variational inference. For both experiments, we create prediction sets in the same way. Given an error tolerance $\alpha$, for each method (stochastic gradient descent, deep ensembles, and mean-field variational inference) and each evaluation dataset, we produce predicted probabilities $\hat{\boldsymbol{\pi}}(\mathbf{x})$ and then create...

**Predictive Credible Sets:**  We order the probabilities $\hat{\pi}_i(\mathbf{x}) \in \hat{\boldsymbol{\pi}}(\mathbf{x})$ from greatest to least and continue adding the corresponding labels until the cumulative probability mass just exceeds $1 - \alpha$. We sometimes abbreviate this method as *cred*.

**Threshold Prediction Sets:**  Using a calibration dataset $\mathcal{D}_{\mathrm{cal}}$ taken from the in-distribution test set, we compute scores for each example using the score function (Sadinle et al., 2019):

$$s(\mathbf{x}, y) = 1 - \hat{\pi}_y(\mathbf{x}).$$

Then we take the

$$[(1 - \alpha)(1 + \frac{1}{|\mathcal{D}_{\mathrm{cal}}|})]\text{-quantile}$$

of these scores which we call $\tau$. Then for each input we want to evaluate for, we create prediction sets as

$$\mathbb{Y}(\mathbf{x}) := \{y \in \boldsymbol{\mathcal{Y}} \mid s(\mathbf{x}, y) \leq \tau\}$$

where $\boldsymbol{\mathcal{Y}}$ is the sample space for labels $y$. We sometimes abbreviate this method as *thr*.

**Adaptive Prediction Sets:** Using a calibration dataset $\mathcal{D}_{\text{cal}}$ taken from the in-distribution test set, we compute scores for each example using the score function

$$s(\mathbf{x}, y) = \hat{\pi}_1(x) + \cdots + U\hat{\pi}_y(x),$$

where $\hat{\pi}_1(x) \geq \cdots \geq \hat{\pi}_y(x)$ and $U$ is a uniform random variable in $[0, 1]$ to break ties (Romano et al., 2020). As in the case of threshold prediction, we take the

$$[(1 - \alpha)(1 + \frac{1}{|\mathcal{D}_{\text{cal}}|})]\text{-quantile}$$

of these scores which we call $\tau$. Then for each input we want to evaluate for, we create prediction sets as

$$\mathbb{Y}(\mathbf{x}) := \{y \in \boldsymbol{\mathcal{Y}} \mid s(\mathbf{x}, y) \leq \tau\}$$

where $\boldsymbol{\mathcal{Y}}$ is the sample space for labels $y$. We sometimes abbreviate this method as *aps*.

**Remark on Adaptive Prediction Sets**: It is useful to note that for coverage to be tight (i.e. having the upper bound in equation (4) of the paper), adaptive prediction sets requires distinct conformity scores. To handle this, an additional standard uniform random variable is used. During the calibration phase, we take $|\mathcal{D}_{\text{cal}}|$ random samples from this variable and subtract it from the scores before computing the quantile $\tau$. And during the prediction phase, we take $|\mathcal{D}_{\text{test}}|$ random samples and subtract it from the scores before checking if they are less than or equal to $\tau$, where $\mathcal{D}_{\text{test}}$ is the test set. We use the implementation from Stutz et al. (2021), which allows one to input a random seed to do the above procedure.

### B.1 CIFAR10 and CIFAR10-Corrupted

In the CIFAR10 experiment we evaluate the prediction sets on (i) the CIFAR10 test set (Krizhevsky et al., 2009), and (ii) all CIFAR10-Corrupted test sets (Hendrycks and Dietterich, 2019). The CIFAR10-Corrupted test sets contain 19 different corruptions, each with intensities ranging from 1 to 5.

The $5 \times 19 = 95$ CIFAR10-Corrupted test sets are all corrupted versions of the original CIFAR10 test set. Thus, when we take the $1,000$ examples from the original CIFAR10 test set to use as a calibration dataset for our conformal methods, we also take the $1,000$ corresponding (semantically similar) examples from all the CIFAR10-Corrupted test sets. We evaluate the three prediction set methods on the remaining $9,000$ examples from the original CIFAR10 test set, and then all the trimmed CIFAR10-Corrupted test sets (which each contain $9,000$ examples). We run the procedure of taking a calibration dataset, finding $\tau$, and computing prediction sets on all the test sets with three different seeds $\{1, 2, 3\}$. Then we take the average accuracy, marginal coverage, and set size across these three seeds. The accuracy results are presented in the main paper. The accuracy, marginal coverage, and set size results for each dataset are presented in section F.2. In the main paper we go further and take the average marginal coverage and average set size *across* data sets at each intensity and report those summarized results.

Table 5: Different type of corruptions in CIFAR10-Corrupted.

|   | Corruption Type |
|---|---|
| 1 | brightness |
| 2 | contrast |
| 3 | defocus blur |
| 4 | elastic |
| 5 | fog |
| 6 | frost |
| 7 | frosted glass blur |
| 8 | gaussian blur |
| 9 | gaussian noise |
| 10 | impulse noise |
| 11 | jpeg compression |
| 12 | motion blur |
| 13 | pixelate |
| 14 | saturate |
| 15 | shot noise |
| 16 | snow |
| 17 | spatter |
| 18 | speckle noise |
| 19 | zoom blur |

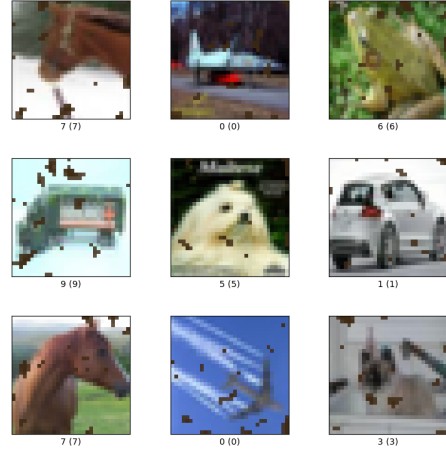

Figure 8: An example of the *spatter* corruption at intensity level 4. https://www.tensorflow.org/datasets/catalog/cifar10_corrupted

## B.2   MedMNIST: organCmnist And organSmnist

In the MedMNIST experiment we only have two test sets. The organ**C**mnist test set (containing 8,268 examples) and the organ**S**mnist test set (containing 8,829 examples). The **C** in organ**C**mnist standards for *coronal* and the **S** in organ**S**mnist stands for *sagittal* (see Figure 7). We take 500 examples from the organ**C**mnist test set to use as a calibration dataset for our conformal methods. We then evaluate the three prediction set methods on the remaining examples from the organ**C**mnist test set as well as on the organ**S**mnist test set, the latter serving as our out-of-distribution test set. We run the procedure of taking a calibration dataset, finding $\tau$, and computing prediction sets on all the test sets with three different seeds $\{1, 2, 3\}$. Then we take the average accuracy, marginal coverage, and set size across these three seeds. The results are presented in the main paper. The class proportions for all splits of both organ**C**mnist and organ**S**mnist are shown in Figure 9.

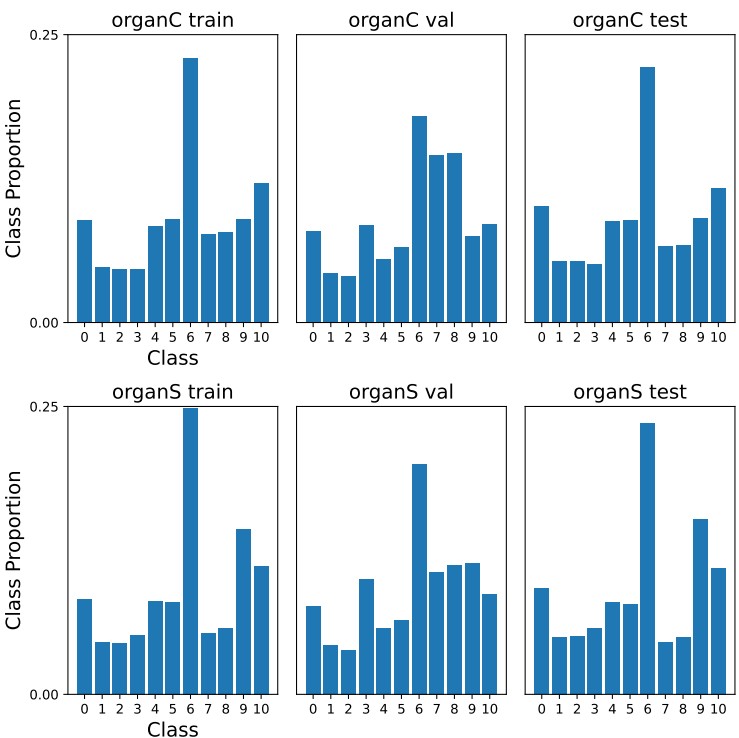

Figure 9: Class propotions for both the organ**C**mnist datasets and the organ**S**mnist datasets.

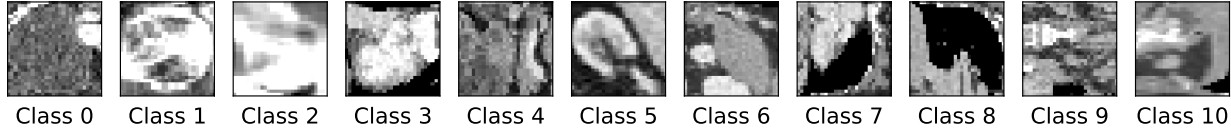

Figure 10: An example image from each of the 11 classes in the organ**C**mnist dataset.

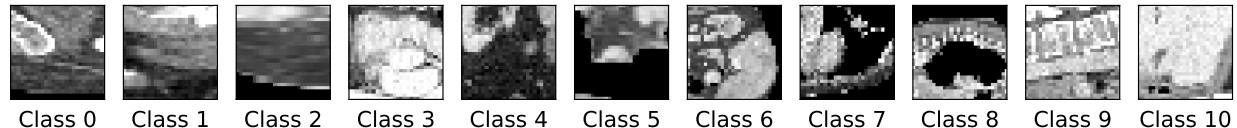

Figure 11: An example image from each of the 11 classes in the organ**S**mnist dataset.

## C SPACE & COMPLEXITY

**Space:** If $n$ is the number of learnable parameters for SGD, then MFV requires $2n$ learnable parameters. This is due to treating each weight as a random variable from a Gaussian distribution, and instead having to fit the two parameters governing that distribution. If $k$ is the number of models in the ensemble, then ENS requires $kn$ learnable parameters. We use $k = 5$. If $q$ is the number of samples collected, then SGHMC requires $qn$ parameters to be kept in order to run inferences.

**Runtime:** If $m$ is the number of forward passes needed to predict using SGD, then MFV inference requires $pm$ forward passes where $p$ is the number of samples to construct a Monte Carlo approximation for the Bayesian model average. During training we have $p = 1$ and during evaluation we have $p = 30$. If $k$ is the number of models in the ensemble, then deep ensembles requires $kn$ forward passes. We use $k = 5$. If $q$ is the number of samples collected for SGHMC then inference requires $qm$ forward passes.

The Laplace approximation we use is post-hoc and needs to be "fit" to the training data (i.e. the Hessian factors need to be computed which require the data). We defer to Appendix B of Daxberger et al. (2021) for space and runtime details.

## D  SOFTWARE PACKAGES

- Python 3, PSF License Agreement (Van Rossum and Drake, 2009).

- Matplotlib, Matplotlib License Agreement (Hunter, 2007).

- Seaborn, BSD License (Waskom, 2021).

- Numpy, BSD License (Harris et al., 2020).

- JAX, Apache 2.0 License (Bradbury et al., 2018).

- Haiku, Apache 2.0 License (Hennigan et al., 2020).

- Tensorflow Datasets, Apache 2.0 License (TFD).

- google-research/bnn_hmc, Apache 2.0 License (Izmailov et al., 2021).

- google-deepmind/conformal_training, Apache 2.0 License (Stutz et al., 2021).

- aleximmer/Laplace, MIT License (Daxberger et al., 2021).

## E  COMPUTE

We ran our experiments on an Ubuntu 18.04.6 system with a dual core 2.10GHz processor and 754 GiB of RAM. We also used a single Tesla V100-SXM2 GPU with 32 GiB of RAM.

**CIFAR10 Experiment**  For *training*, SGD takes $\approx 3.6$ minutes per seed, MFV takes $\approx 4.6$ minutes per seed, SGHMC takes $\approx 4$ hours, and LAPLACE takes $\approx 5$ minutes. For *evaluation*, SGD takes $\approx .3$ minutes per seed, MFV takes $\approx .3$ minutes per seed, ENS takes $\approx .4$ minutes per seed, SGHMC takes $\approx 20$ minutes per seed, and LAPLACE takes $\approx 5$ minutes per seed. Assuming (i) you train SGD and MFV with 5 different seeds, (ii) you evaluate using 3 seeds: the approximate time to run the CIFAR10 experiment is

$$(\underbrace{(3.6 + 4.6) \times 5 + 240 + 5)}_{\text{training}} + (\underbrace{(0.3 + 0.3 + 0.4 + 20 + 5) \times 3}_{\text{evaluation}} \times \underbrace{96}_{\#\text{ datasets}}) \approx 129.5 \text{ hours}$$

**MedMNIST Experiment**  For *training*, SGD takes $\approx 2.6$ minutes per seed, MFV takes $\approx 5$ minutes per seed, and SGHMC takes $\approx 5$ hours. For evaluation, SGD takes $\approx .4$ minutes per seed, MFV takes $\approx .8$ minutes per seed, ENS takes $\approx .7$ minutes per seed, SGHMC takes $\approx 30$ minutes per seed, and LAPLACE takes $\approx 5$ minutes per seed. Assuming (i) you train SGD and MFV with 5 different seeds, (ii) you evaluate using 3 seeds: the approximate time to run the MedMNIST experiment is

$$(\underbrace{(2.6 + 5) \times 5 + 300)}_{\text{training}} + (\underbrace{(0.4 + 0.8 + 0.7 + 30) \times 3}_{\text{evaluation}} \times \underbrace{2}_{\#\text{ datasets}}) \approx 7.25 \text{ hours}$$

## F   ADDITIONAL EXPERIMENTAL RESULTS

### F.1   CIFAR10-Corrupted Box Plot Results

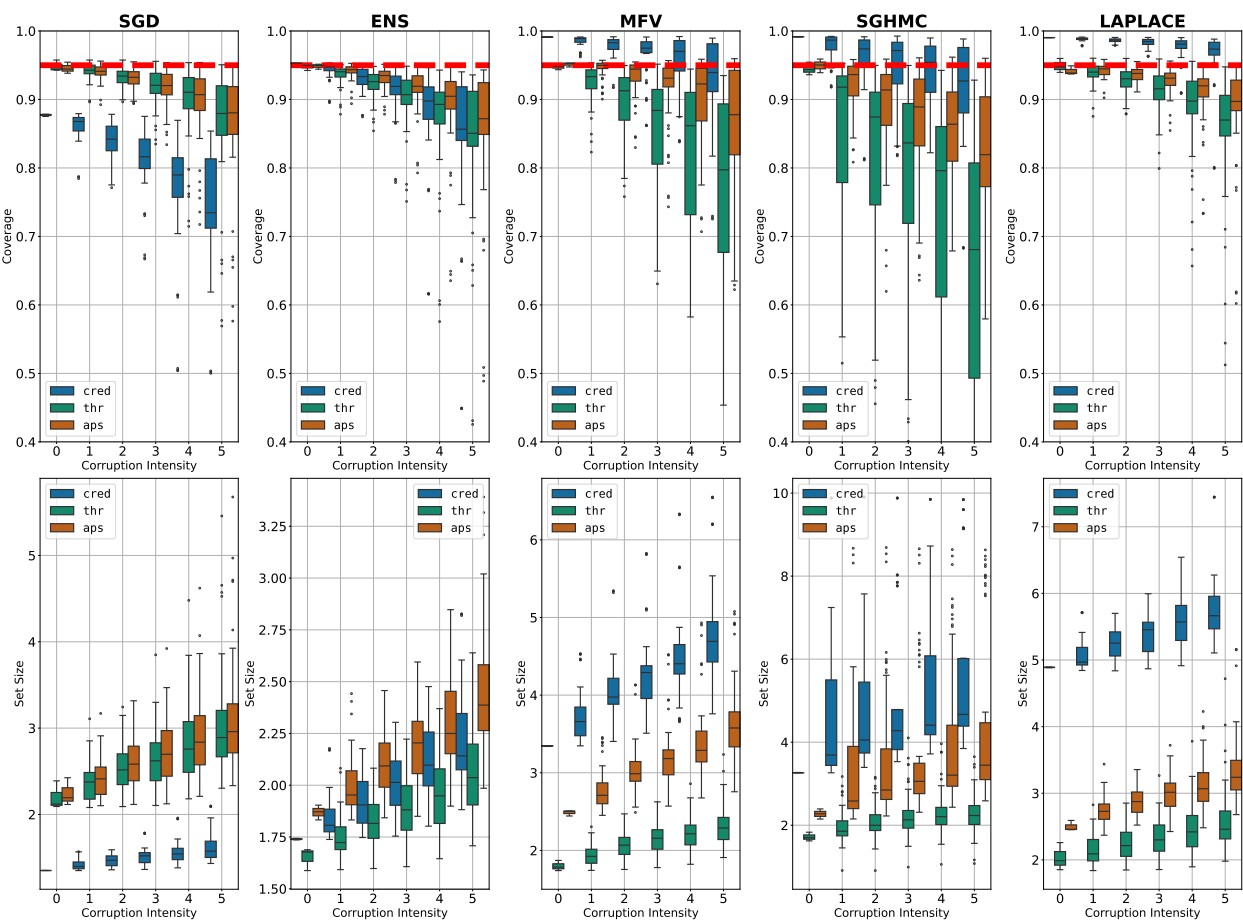

Figure 12: Box plot results for the CIFAR10 experiment with an error tolerance of 0.05. Note that some outliers occur that are not shown for coverage for ease of visualization.

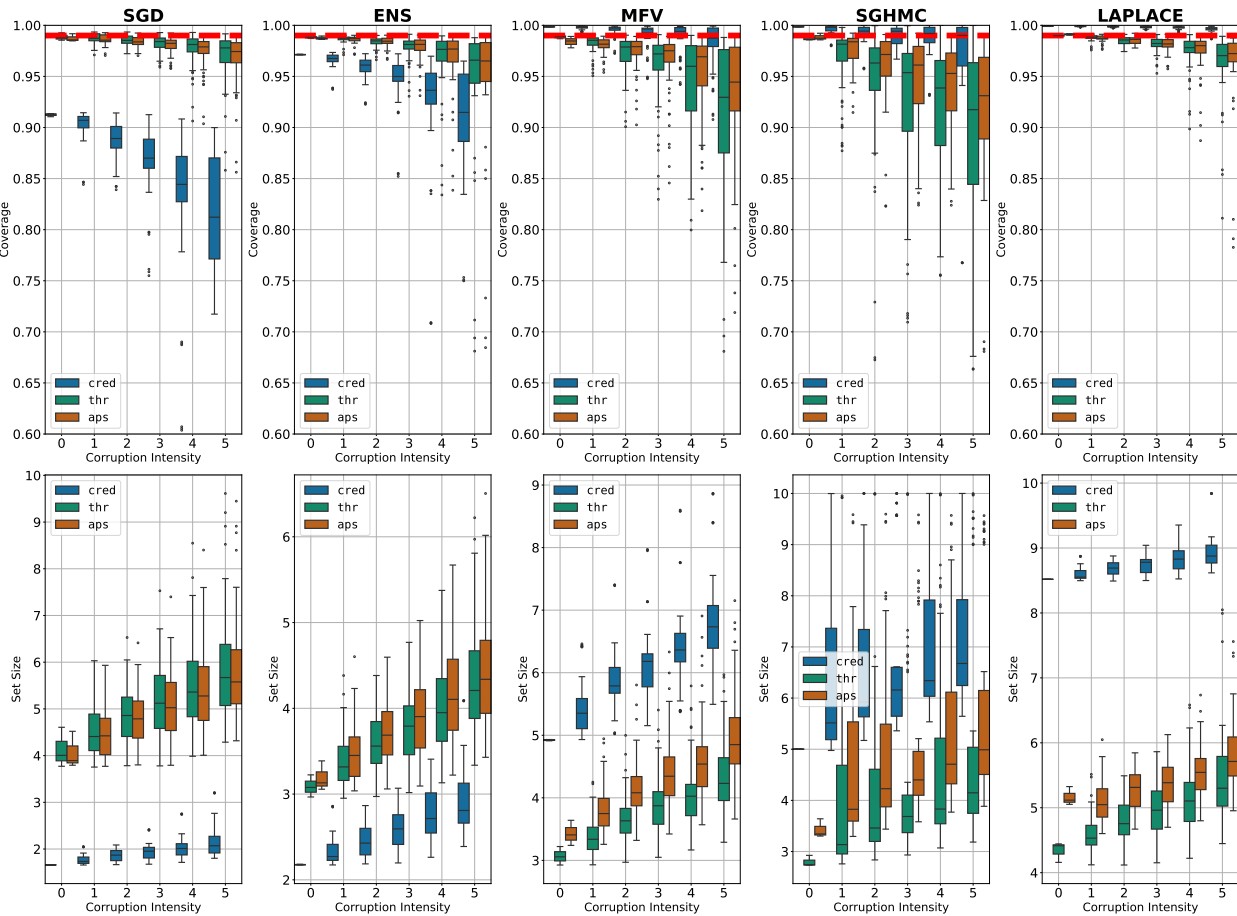

Figure 13: Box plot results for the CIFAR10 experiment with an error tolerance of 0.05. Note that some outliers occur that are not shown for coverage for ease of visualization.

### F.2 CIFAR10-Corrupted Per-Dataset Results

### 0.05 Error Tolerance

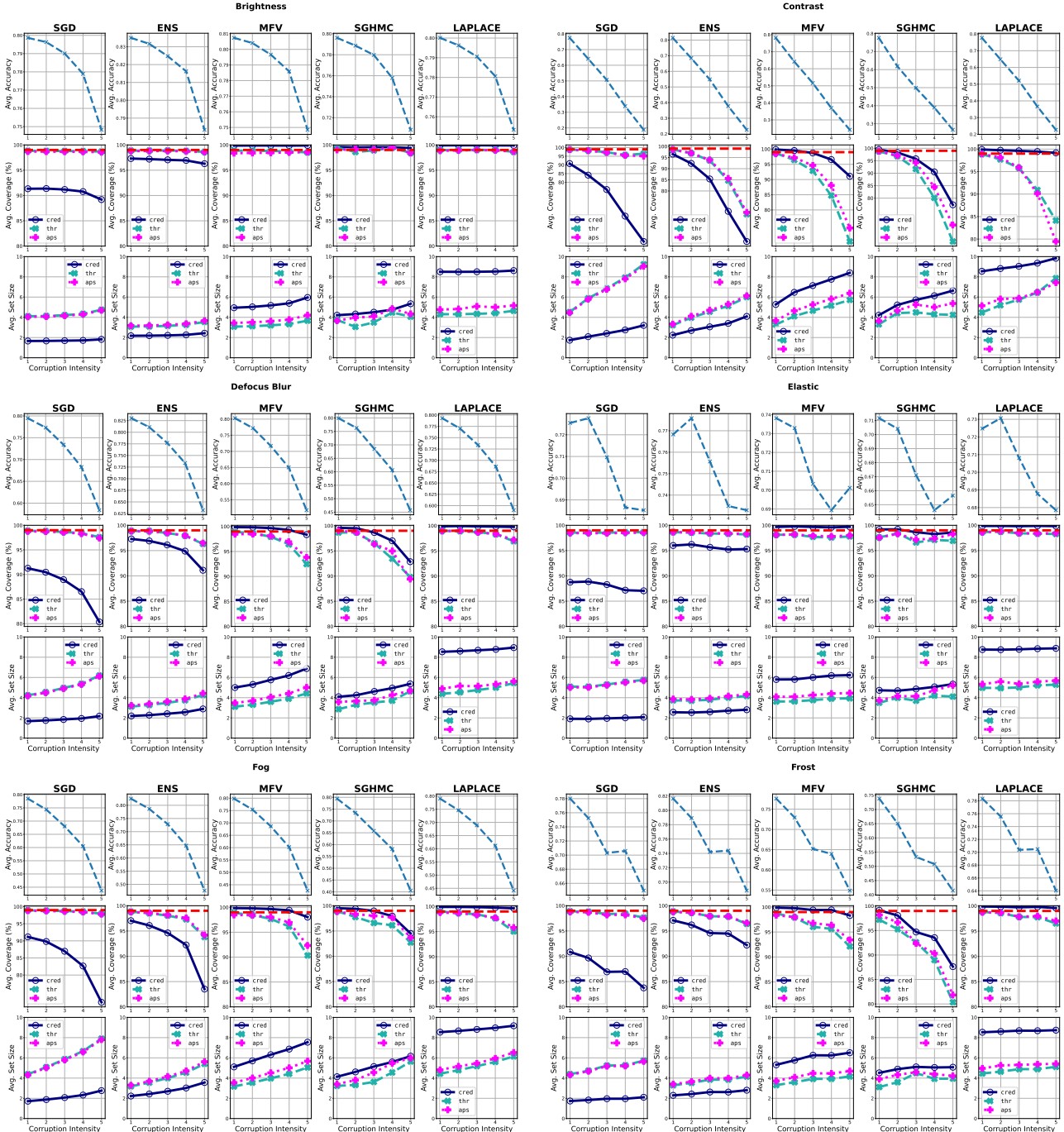

Figure 14: CIFAR10-Corrupted per-dataset results at the 0.05 Error Tolerance.

## 0.05 Error Tolerance

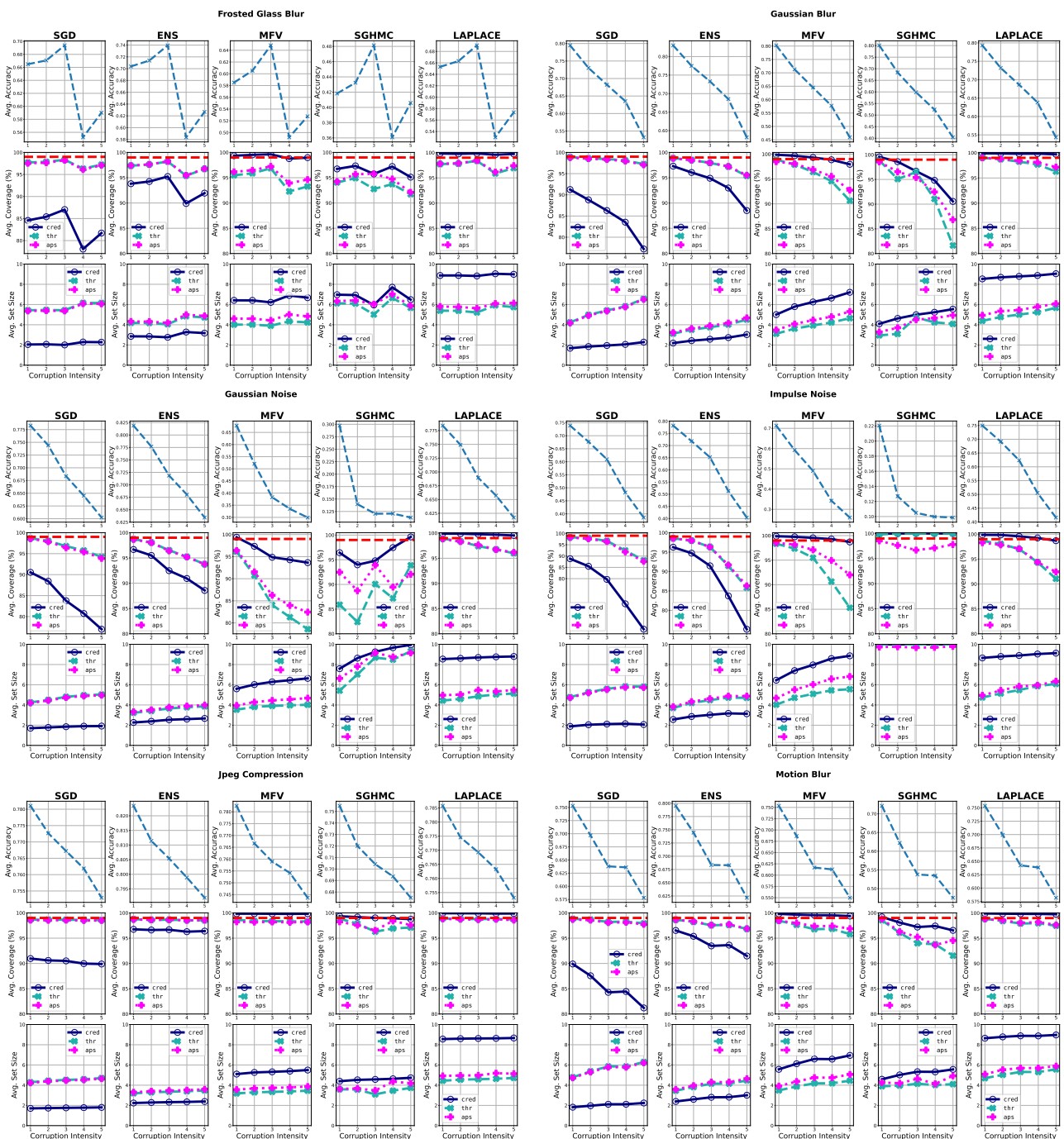

Figure 15: CIFAR10-Corrupted per-dataset results at the 0.05 Error Tolerance.

**0.05 Error Tolerance**

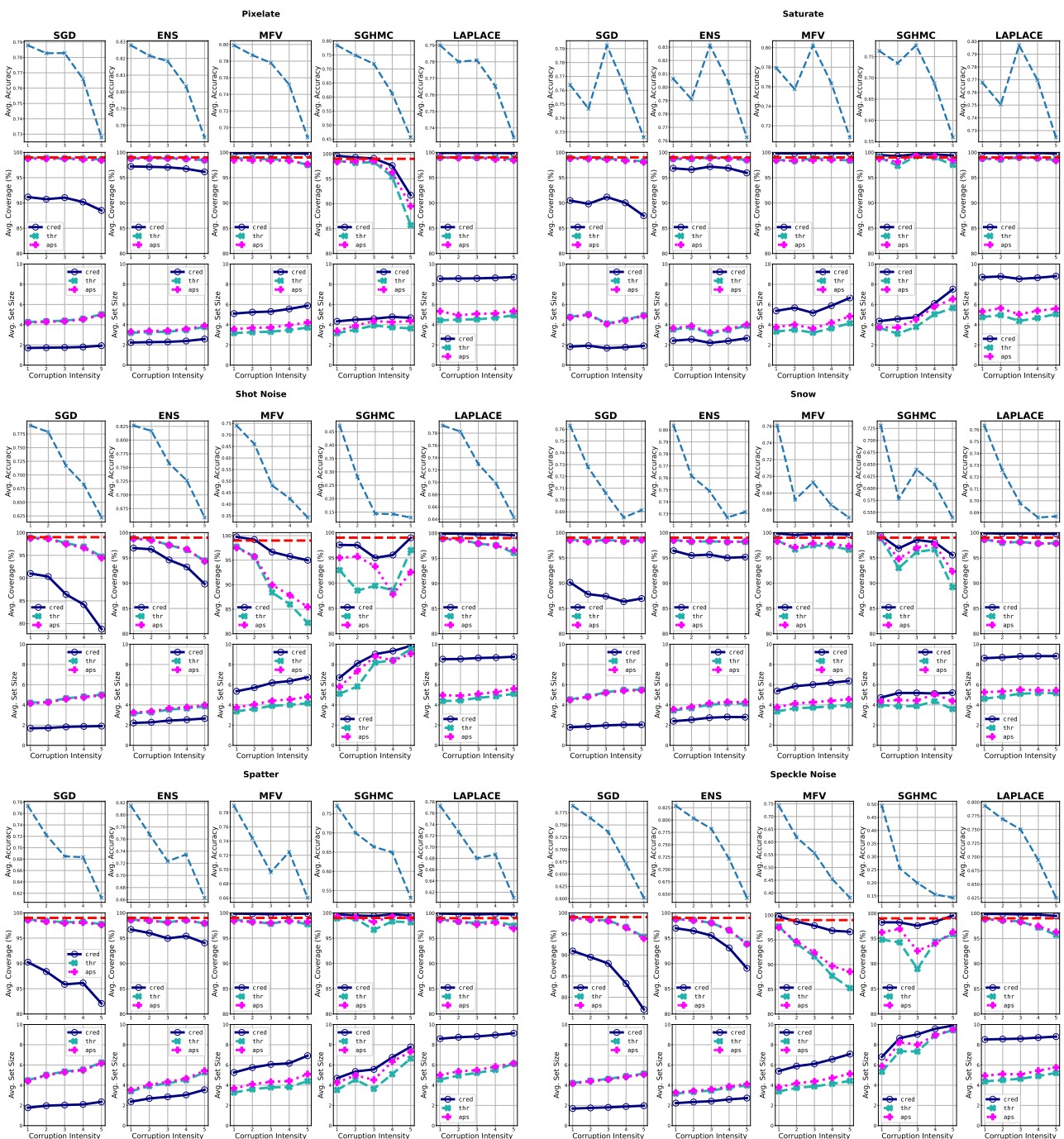

Figure 16: CIFAR10-Corrupted per-dataset results at the 0.05 Error Tolerance.

## 0.05 Error Tolerance

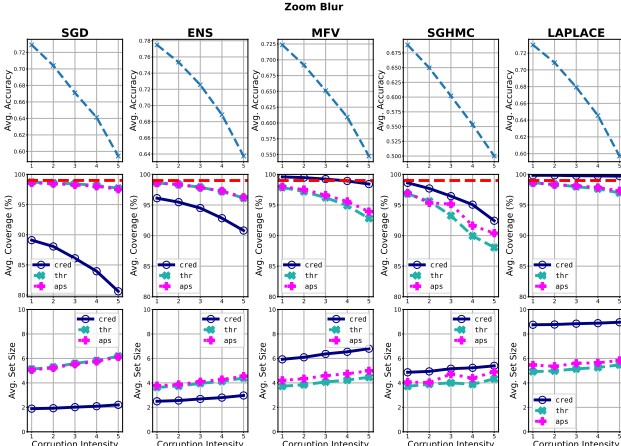

Figure 17: CIFAR10-Corrupted per-dataset results at the 0.05 Error Tolerance.

**0.01 Error Tolerance**

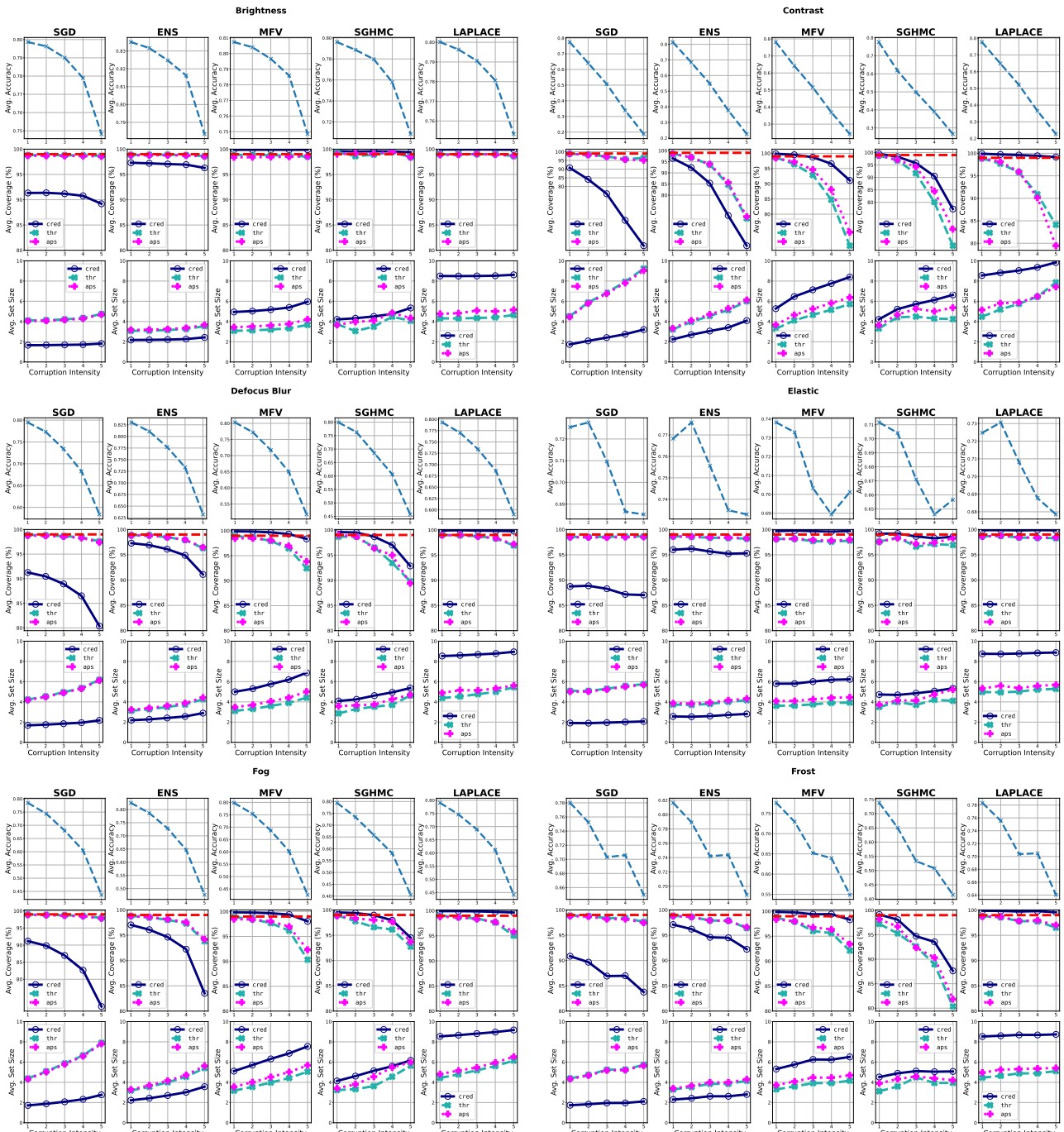

Figure 18: CIFAR10-Corrupted per-dataset results at the 0.01 Error Tolerance.

**0.01 Error Tolerance**

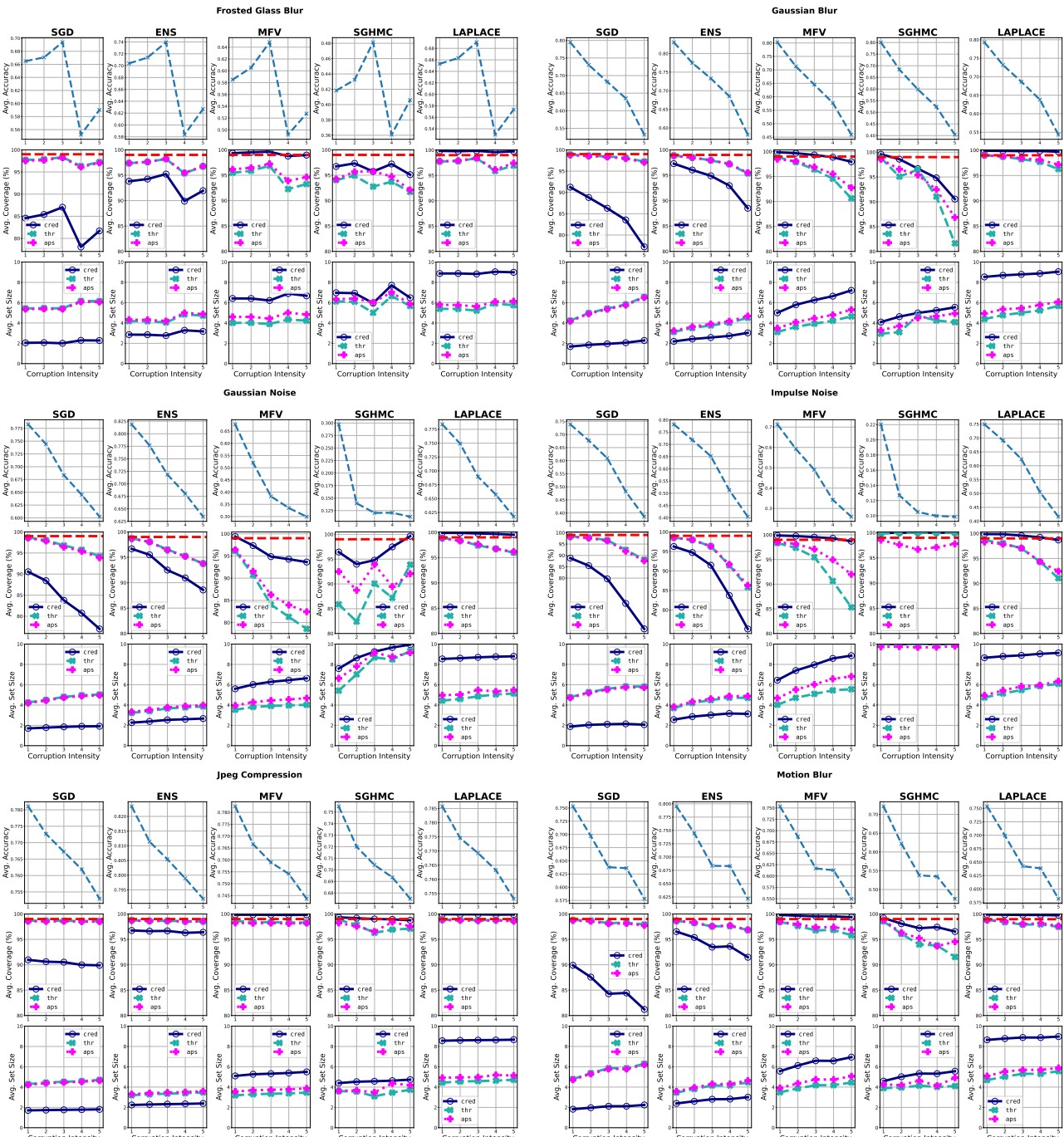

Figure 19: CIFAR10-Corrupted per-dataset results at the 0.01 Error Tolerance.

## 0.01 Error Tolerance

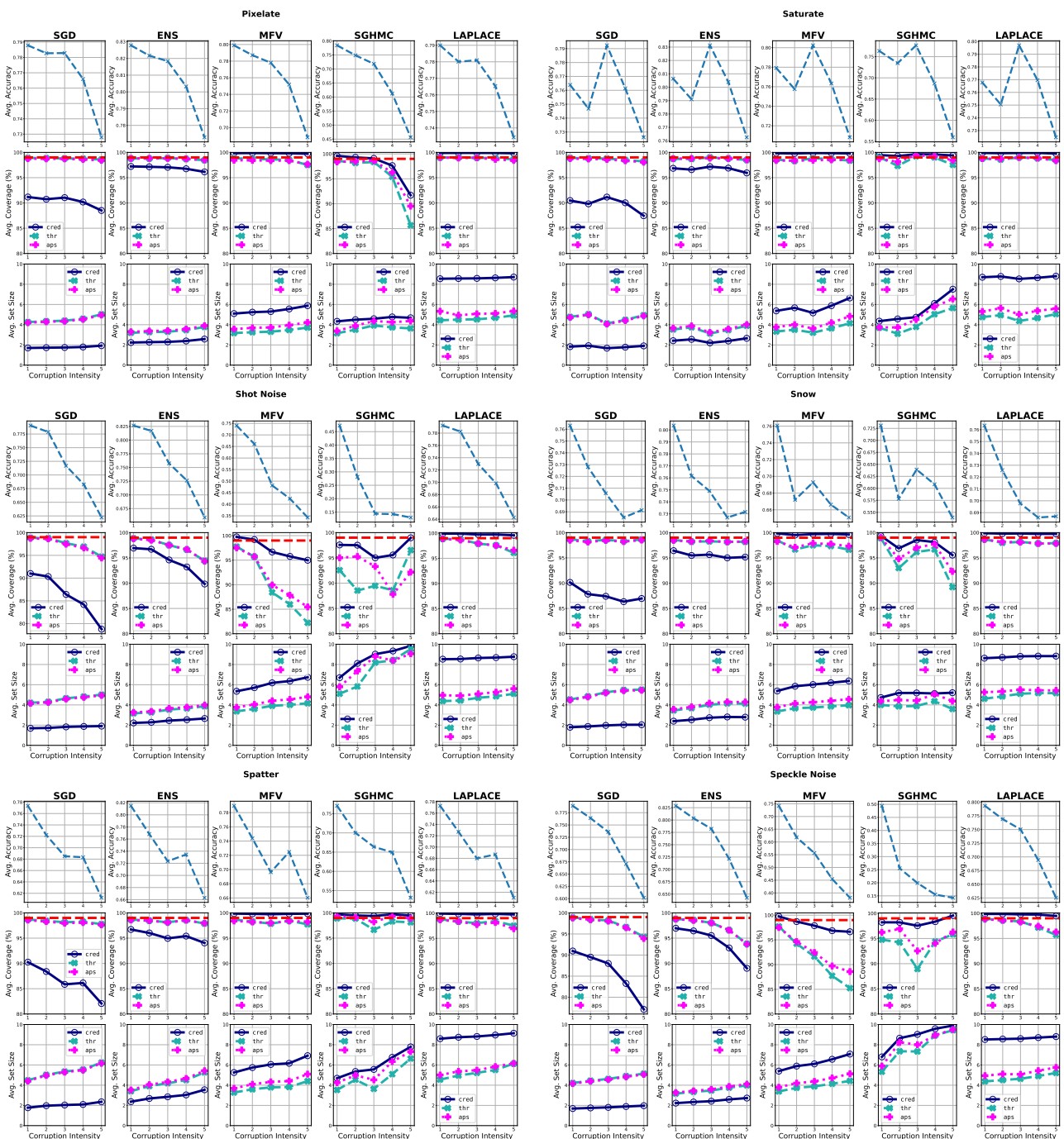

Figure 20: CIFAR10-Corrupted per-dataset results at the 0.01 Error Tolerance.

## 0.01 Error Tolerance

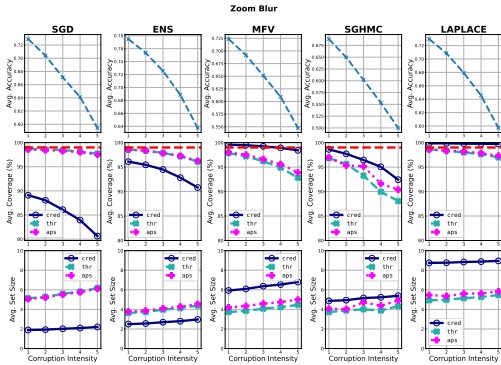

Figure 21: CIFAR10-Corrupted per-dataset results at the 0.01 Error Tolerance.

## F.3 MedMNIST Results With Variance Over Calibration / Test Splits

| Error | Dataset | Train Method / Pred-Set Method | SGD | ENS | MFV | SGHMC |
|---|---|---|---|---|---|---|
| **0.05** | organ**C**mnist | Credible (*cred*) | $94.98 \pm 0.0005$ | $97.02 \pm 0.0005$ | $99.21 \pm 0.0$ | $98.92 \pm 0.0$ |
| | | Threshold (*thr*) | $95.37 \pm 0.011$ | $95.41 \pm 0.0076$ | $94.64 \pm 0.0068$ | $94.85 \pm 0.0$ |
| | | Adaptive (*aps*) | $95.67 \pm 0.011$ | $95.77 \pm 0.0124$ | $94.61 \pm 0.0038$ | $95.00 \pm 0.0083$ |
| | organ**S**mnist | Credible (*cred*) | $68.44 \pm 0.0$ | $76.52 \pm 0.0$ | $92.16 \pm 0.0$ | $86.27 \pm 0.0$ |
| | | Threshold (*thr*) | $69.65 \pm 0.052$ | $64.86 \pm 0.035$ | $66.46 \pm 0.024$ | $64.70 \pm 0.0$ |
| | | Adaptive (*aps*) | $77.63 \pm 0.026$ | $80.10 \pm 0.0329$ | $84.21 \pm 0.006$ | $80.14 \pm 0.004$ |
| **0.01** | organ**C**mnist | Credible (*cred*) | $97.46 \pm 0.0004$ | $98.85 \pm 0.0005$ | $99.85 \pm 0.0003$ | $99.59 \pm 0.0$ |
| | | Threshold (*thr*) | $99.06 \pm 0.004$ | $99.13 \pm 0.0073$ | $98.93 \pm 0.0068$ | $98.88 \pm 0.0$ |
| | | Adaptive (*aps*) | $99.43 \pm 0.002$ | $99.37 \pm 0.0052$ | $99.41 \pm 0.0020$ | $99.15 \pm 0.0024$ |
| | organ**S**mnist | Credible (*cred*) | $81.64 \pm 0.0$ | $88.48 \pm 0.0$ | $98.61 \pm 0.0$ | $94.00 \pm 0.0$ |
| | | Threshold (*thr*) | $91.54 \pm 0.03$ | $91.52 \pm 0.05$ | $89.94 \pm 0.05$ | $86.84 \pm 0.0$ |
| | | Adaptive (*aps*) | $95.86 \pm 0.01$ | $95.07 \pm 0.037$ | $96.42 \pm 0.008$ | $92.10 \pm 0.01$ |

Table 6: MedMNIST Coverage with Variances

| Error | Dataset | Train Method / Pred-Set Method | SGD | ENS | MFV | SGHMC |
|---|---|---|---|---|---|---|
| **0.05** | organ**C**mnist | Credible (*cred*) | $1.27 \pm 0.002$ | $1.41 \pm 0.002$ | $2.90 \pm 0.0038$ | $1.87 \pm 0.0$ |
| | | Threshold (*thr*) | $1.31 \pm 0.132$ | $1.18 \pm 0.066$ | $1.21 \pm 0.0501$ | $1.13 \pm 0.0$ |
| | | Adaptive (*aps*) | $1.63 \pm 0.155$ | $1.56 \pm 0.15$ | $1.97 \pm 0.0354$ | $1.56 \pm 0.031$ |
| | organ**S**mnist | Credible (*cred*) | $2.15 \pm 0.0$ | $2.57 \pm 0.0$ | $4.79 \pm 0.0$ | $3.13 \pm 0.0$ |
| | | Threshold (*thr*) | $2.24 \pm 0.589$ | $1.59 \pm 0.212$ | $1.50 \pm 0.122$ | $1.37 \pm 0.0$ |
| | | Adaptive (*aps*) | $3.57 \pm 0.48$ | $3.23 \pm 0.513$ | $3.26 \pm 0.073$ | $2.59 \pm 0.053$ |
| **0.01** | organ**C**mnist | Credible (*cred*) | $1.83 \pm 0.0052$ | $2.13 \pm 0.0047$ | $5.44 \pm 0.0093$ | $2.91 \pm 0.0$ |
| | | Threshold (*thr*) | $3.05 \pm 0.6439$ | $2.75 \pm 0.854$ | $2.61 \pm 0.6606$ | $1.91 \pm 0.0$ |
| | | Adaptive (*aps*) | $4.38 \pm 0.9158$ | $3.82 \pm 1.6073$ | $4.26 \pm 0.3742$ | $2.61 \pm 0.2080$ |
| | organ**S**mnist | Credible (*cred*) | $4.24 \pm 0.0$ | $5.06 \pm 0.0$ | $8.07 \pm 0.0$ | $4.83 \pm 0.0$ |
| | | Threshold (*thr*) | $7.14 \pm 1.05$ | $6.55 \pm 2.30$ | $4.21 \pm 1.15$ | $3.15 \pm 0.0$ |
| | | Adaptive (*aps*) | $8.13 \pm 0.58$ | $7.72 \pm 1.56$ | $6.70 \pm 0.45$ | $4.39 \pm 0.30$ |

Table 7: MedMNIST Set Size with Variances

