# OpenReview forum: "On the Out-of-Distribution Coverage of Combining Split Conformal Prediction and Bayesian Deep Learning"
_TMLR — Accepted by TMLR_

### Review · Reviewer_dkAf · 2023-12-08

**Summary Of Contributions:**

The paper analyses the combined effect of using Bayesian inference for parameter estimation and conformal prediction for calibrating the predictions in the context of neural networks, specifically from the perspective of how well the predictions for out-of-distribution samples behave.

The two techniques are used in sequential manner and the main result of the work is observing that conformal predictions hurt when the original predictions were underconfident and help when they were overconfident. This happens because out-of-distribution samples require higher prediction uncertainty and conformal predictions ignore this but merely try to correct the confidence for the training distribution.

**Audience:**

No

**Claims And Evidence:**

No

**Requested Changes:**

I believe the paper would require substantial changes to warrant publication. It should consider broader range of actual Bayesian DL techniques or alternatively it should not be presented from the perspective of Bayesian DL at all. More importantly, I feel that the overall approach of merely running standard conformal prediction as postprocessing for Bayesian predictions is not scientifically interesting, and to fix this the authors would need to go deeper into how out-of-distribution samples should be handled in this setting. The current results would serve as good baselines.

Minor remarks for polishing:
- The evaluation measures should be provided more formally, in form of equations
- The Related work section would work better a bit later in the paper, when the reader knows the methods (Sections 2.2-2.4) better
- Fig 4(a) and 5 are sightly confusing; these are not line plots and the linear visual style misleads the reader, even though I understand the attempt to make the visually similar to Fig 2 and 3

**Strengths And Weaknesses:**

Combining Bayesian inference for accounting uncertainty and conformal predictions for ensuring in-distribution calibration is an intriguing possibility. However, the present work is a rather lightweight treatment of this intersection, since the authors simply apply the techniques in a sequential manner: Standard conformal prediction technique is used on top of predictions for Bayesian neural networks. The specific research question is how the combination works for out-of-distribution samples.

My first problem with this is that the approach is scientifically questionable. As the authors clearly say, the guarantees of conformal predictions do not hold for out-of-distribution samples and here the authors measure their quality specifically in this setting. Since conformal predictions are used as postprocessing there is no reason to believe the predictions would be good for out-of-distribution samples irrespective of how the initial predictions were formed. In other words, I do not see a clear theoretical justification for the approach. I understand the work is framed from the perspective of evaluating how this would work, rather than proposing it as a new approach, but I would still expect the approach being evaluated to be one that has some justification for why it could work. Why not consider conformal predictions methods that attempt to account for the out-of-distribution samples? The authors already cite works that present such extensions, such as Barber et al. (2023).

The main empirical result follows from known properties of conformal predictions for out-of-distribution samples. By definition, we need higher prediction uncertainty to model samples that do not follow the training distribution. Conformal prediction methods form sets that capture uncertainty well for in-distribution samples, and hence by definition will improve coverage of out-of-distribution samples when the initial predictions were overconfident and degrade it when the initial predictions were underconfident. The authors obverse this to be the case and provide the intuitive explanations, but the observations are as everyone would expect. For instance, SGD is obviously overconfident and conformal prediction then naturally helps also for out-of-distribution samples. I find it hard to believe this would be a new insight for someone familiar with the topic.

Another problem relates to scoping. Starting from the title, the paper talks about Bayesian deep learning. However, it is very shallow in terms of actual Bayesian DL. One of the so-called BDL methods is standard SGD, the other is an ensemble that was originally proposed specifically as a non-Bayesian alternative for quantifying uncertainty, and the final one is standard mean-field variational approximation that indeed is a BDL method but by no means state-of-the-art. In summary, the authors effectively compare one Bayesian method with two non-Bayesian methods and hence the paper is mispositioned in terms of title and story. In addition, the authors miss an opportunity of evaluating how alternative BDL methods (e.g. Laplace approximation, SGLD, Monte Carlo dropout, flow-based VI etc) would behave in this sense. There are well-known differences between the various approximations in terms of the predictive uncertainty calibration, and empirical quantification of those in conjunction with conformal predictions would provide results that would no longer be obvious and hence would be of interest for some readers.

The experiments are sound, but a bit shallow as the authors only inspect the problem from one perspective. I see no added value of explaining use of the exact same techniques on two different data sets as two separate experiments with detailed verbose explanations. Both experiments answer to the same general question and could well be combined into one that provides the numerical results for not just these two data sets but for a broader range of data sets, so that the observations could be stated in a more general form.

Finally, I would like to commend the authors on a well-prepared Supplement. Explicit listing of software and their licenses is exemplary and I appreciate the detailed analysis of computing resources described in Appendix E.

---

### Review · Reviewer_NvAv · 2023-12-11

**Summary Of Contributions:**

The submission proposes an empirical evaluation of the effects of conformal prediction (CP) calibration techniques on out-of-distribution (OOD) coverage for predictors learned with stochastic gradient descent or Bayesian deep learning techniques. Experiments are carried on the CIFAR10 and CIFAR10-Corrupted as well as on the organCmnist and organSmnist datasets. Results show that depending on the choice of model, calibration technique, and desired miscoverage level, CP can reduce out-of-distribution coverage.

**Audience:**

Yes

**Claims And Evidence:**

No

**Requested Changes:**

1. Off-the-shelf CP for OOD calibration

I recognize the practical importance of showing the effects of CP on OOD calibration for Bayesian deep learning approaches. However, this is not the intended use of the CP methods used in the paper. Is the message here that practitioners should be careful when using off-the-shelf CP to provide claims on the reliability of their models? Using a calibration set drawn from a different distribution does not guarantee coverage, and that is okay.

In Sec. 3.1, the sentence "However, this could [...] more confident than it *should* be." is confusing because CP yields a predictor that is as confident as a user asks it to be given a calibration set. If the calibration set does not represent the distribution the model is supposed to work on, and we do not know anything about this distribution, then there is little to hope for. Maybe a more appropriate approach would be to use online CP methods that adapt over time?

Similarly, in the Conclusion paragraph of Sec. 5, I am not sure the claim that "conformal prediction can cause unintended consequences" reflects that the CP methods used in the paper are not intended for OOD calibration.

2. Intuitive explanation

I agree that if I am overconfident (i.e., I undercover), CP will yield a larger prediction set. On the other hand, if I am underconfident (i.e., I overcover), CP will yield a smaller prediction set. But still, this applies to the calibration set.

I am not sure I follow how the experimental results show this is the underlying mechanism that drives OOD coverage. The figures in the paper seem to show the average set size tends to increase with the level of corruption, which is not immediately obvious to me. There may be OOD distributions where the model is very confident in the wrong answer, and the average set size may not change? Is this phenomenon related to the specific models and datasets used in the experiments?

The intuitive explanation for why CP may hurt OOD calibration sounds a little like "the trivial set predictor that includes all possible labels provides 100% OOD coverage".

Am I misunderstanding the story here? I am looking forward to hearing the authors' perspective.

3. Miscellaneous comments:

* Typo on page 5: "overly confident in *its* predictions" -> "overly confident in *their* predictions"
* Eq. (6): what does the $\overset{(i)}{=}$ symbol mean?
* Fig. 3: Could the "Corruption intensity" labels be repeated across columns?
* First paragraph of Sec. 4: "that were produced for every single corruped CIFAR10 dataset". Could this sentence be rephrased to "that were produced for every single CIFAR10-Corruped dataset" to clarify this is an existing dataset?
* Claim at the bottom of page 7: "For example, if one is looking for [...]". Deep ensembles with *aps* at the 0.01 error tolerance do not always provide coverage greater than 96%? The line crosses 95% at corruption intensity 5. The average set size seems to be slightly larger than deep ensembles with *thr*? Both lines seems to overlap significantly and they are hard to distinguish.

**Strengths And Weaknesses:**

Strengths:
1. Clarity of presentation: the paper is well written.
2. Motivation: the interaction of conformal prediction with Bayesian techniques is relevant and important.
3. Practicality: the results presented in the paper may guide practitioners to a more responsible use of conformal prediction in real-world scenarios.

Weaknesses:
1. Off-the-shelf CP is not designed for OOD calibration, so why should it to work?
2. I am not sure I understand how the intuitive explanation provided in Sec. 3.1 applies to OOD calibration.

I will expand on these questions and I am looking forward to discussing with the authors to better understand their intention and whether I am missing parts of the story.

---

### Review · Reviewer_JLk9 · 2023-12-30

**Summary Of Contributions:**

The authors conduct an empirical study of the behavior of conformal prediction methods when the usual assumptions required for its theoretical guarantees are broken, such as when there is an unknown distribution shift between training and test sets. In the paper, the authors consider image classification on CIFAR-10 and medical MNIST with neural networks and apply conformal prediction techniques to:
1. Neural networks optimized using a MAP objective with SGD.
2. Deep ensembles.
3. Mean-field (MF) variational Bayesian neural networks.

The authors compare the coverage of conformal predictive sets against Bayesian credible sets. Their experiments show that the credible set approach using the MF approximation can significantly outperform any conformal prediction approach.

**Audience:**

Yes

**Claims And Evidence:**

Yes

**Requested Changes:**

The authors should fix the errors I describe in the weaknesses section. I would be delighted if the authors added more discussion regarding potential fixes or added more experiments, but these are not critical.

**Strengths And Weaknesses:**

## Strengths
The authors consider a very relevant question: does conformal prediction perform well in realistic scenarios? While they do not conclusively answer this question in the negative, they demonstrate some fairly simple scenarios where conformal prediction fails and where simple Bayesian credible sets obtained from variational Bayesian neural networks outperform it.

The paper is generally well-written, and the authors' arguments are easy to follow. In particular, the authors explain the core concepts well, such as calibration and conformal prediction; the problem they study is well-motivated, and the presentation and intuitive explanation of the observed phenomena is clear. Furthermore, the figures are very nicely done.

I also looked at the appendix, which is very nicely laid out and seems to provide a lot of additional detail for the theory and the experiments, thus leaving little doubt about the replicability of the paper.

## Weaknesses
Though the authors acknowledge it, the paper's main weakness is its scope seems quite limited.

While the authors diagnose and explain the issue well, they don't offer much of a remedy in the paper. In particular, I don't have much expertise in conformal prediction; my expertise is aligned a lot closer with Bayesian inference, and thus, my takeaway from the paper is essentially that conformal prediction methods are currently not worth the extra computational effort when the simple MFVI method with credible sets outperforms them. Hence, it would be good if the authors could at least discuss potential ways to fix the issues of conformal prediction when the necessary assumptions for its guarantees are broken.

Besides this, there are some smaller issues in the main text:
- The notation $\hat{p}(x)$ is a bit confusing, as it conflicts with the probability notation later; perhaps consider using $\pi(x)$ instead.
- the index of $p_i$ in the first equation in section 2.2 should be $p_k$
- the third point in the list in section 2.3 doesn't seem to be an assumption
- eq 5 is missing a $dw$ term
- $p(w | D) \approx w_{MAP}$ should be $p(w | D) \approx \delta(w - w_{MAP})$
- in the argmax, please use curly braces, as it is a function that selects the element of a set.
- eq 7: "kl between variational and prior" should read "variational posterior"
- footnote 2: "knowledge is known"
- The color assignment between Figs 2a and 2b is inconsistent; please ensure the same method is assigned the same color in both plots.
- In the Figures, what is the "average"? Is it a mean or median? Could the authors show error bars?

## Questions
I am quite surprised at how well MFVI performs; I would be interested in how well a Laplace-approximated BNN (see e.g., [1]), a more state-of-the-art approximate Bayesian model, would perform.

## References
[1] Immer, A., Korzepa, M., & Bauer, M. (2021, March). Improving predictions of Bayesian neural nets via local linearization. In International conference on artificial intelligence and statistics.

---

### Decision · Action_Editor_WGmU · 2024-02-05

**Recommendation:** Accept with minor revision

**Comment:**

After a lively discussion with the reviewers, it seems that the paper is generally interesting enough for the TMLR audience, but that the claims should be slightly modified to make sure that everyone feels confident that they are supported by evidence. Especially based on comments from reviewer NvAv, it is suggested that the claims of the paper be rephrased along the following lines:

- Clearly state and remark that the proposed explanation follows by definition of marginal coverage. For a fixed model, the predictive set with highest chances of increasing OOD coverage will always have largest average set size. According to the proposed explanation, for a fixed model, increasing the chances of OOD coverage without any knowledge on the test distribution will always be at the expense of average set size. This applies to Bayesian DL as well as CP, and not the latter only.

- Remark that across all experiments, the combination of model and prediction set that provides the highest OOD coverage is always the "least confident" (in terms of average set size), with average set sizes that are larger than half of the total possible classes. It is true that these choices will increase the chances of OOD coverage, at the same time, it needs to be acknowledged that a model that predicts more than half of the possible classes is of questionable utility.

- It is stated several times across the manuscript that the main motivation is to quantify the extent to which CP hurts OOD coverage compared to credible predictive sets. Besides including the bar plots in Fig. 4,6, these quantitative findings are not discussed in the text. In my discussion with the authors, it was mentioned "for example, for ~1.25 decrease in average set size, CP with THR causes SGHMC coverage to go down by around 30% (see Figure 3).". This analysis is missing from the current version of the manuscript, and it should be included, in particular with respects to the correlation between differences in OOD coverage and in average set size.

Since these all seem like minor modifications, I am happy to accept with minor revision and check the camera-ready version to make sure the claims have been adjusted.

**Audience:**

The reviewers agree that the general question of the paper, while somewhat naive, is interesting to the TMLR community and that sharing the results in this paper will be useful to the discourse.

**Claims And Evidence:**

There seems to be a perceived minor mismatch between claims and evidence, which should be easy to remedy with a minor revision of the claims. See comment below for details.